# Chemical discrimination of the particulate and gas phases of miniCAST exhausts using a two-filter collection method

Linh Dan Ngo[1,2], Dumitru Duca[1], Yvain Carpentier[1], Jennifer A. Noble[1,*], Raouf Ikhenazene[1], Marin Vojkovic[1], Cornelia Irimiea[3], Ismael K. Ortega[3], Guillaume Lefevre[4], Jérôme Yon[4], Alessandro Faccinetto[2], Eric Therssen[2], Michael Ziskind[1], Bertrand Chazallon[1], Claire Pirim[1], Cristian Focsa[1]

[1]Univ. Lille, CNRS, UMR 8523 – PhLAM – Laboratoire de Physique des Lasers Atomes et Molécules, F-59000 Lille, France
[2]Univ. Lille, CNRS, UMR 8522 – PC2A – Physicochimie des Processus de Combustion et de l'Atmosphère, F-59000 Lille, France
[3]ONERA – The French Aerospace Laboratory, F-91123 Palaiseau, France
[4]Normandie Univ., INSA Rouen, UNIROUEN, CNRS, CORIA, 76000 Rouen, France
[*]now at: CNRS, Aix Marseille Université, PIIM, UMR 7345, 13397 Marseille cedex, France

*Correspondence to*: Yvain Carpentier (yvain.carpentier@univ-lille.fr)

**Abstract.** Combustion of hydrocarbons produces both particulate and gas phase emissions responsible for major impacts on atmospheric chemistry and human health. Ascertaining the impact of these emissions, especially on human health, is not straightforward because of our relatively poor knowledge of how chemical compounds are partitioned between the particle and gas phases. Accordingly, we propose to couple a two-filter sampling method with a multi-technique analytical approach to fully characterize the particulate and gas phase compositions of combustion by-products. The two-filter sampling method is designed to retain particulate matter (elemental carbon possibly covered in a surface layer of adsorbed molecules) on a first quartz fiber filter while letting the gas phase pass through, and then trap the most volatile components on a second black carbon-covered filter. All samples thus collected are subsequently subjected to a multi-technique analytical protocol involving two-step laser mass spectrometry (L2MS), secondary ion mass spectrometry (SIMS), and micro-Raman spectroscopy. Using the combination of this two-filter sampling/multi-technique approach in conjunction with advanced statistical methods we are able to unravel distinct surface chemical compositions of aerosols generated with different set points of a miniCAST burner. Specifically, we successfully discriminate samples by their volatile, semi-volatile and non-volatile polycyclic aromatic hydrocarbon (PAH) contents and reveal how subtle changes in combustion parameters affect particle surface chemistry.

## 1 Introduction

Particulate matter (PM) produced by incomplete combustion of hydrocarbon-based fuels is often found associated to gas phase compounds that include carbon and nitrogen oxides (CO, $CO_2$, and $NO_X$), along with a volatile fraction of organic species encompassing low-mass polycyclic aromatic hydrocarbons (PAHs). The presence of PAHs in the atmosphere is of great concern due to their carcinogenic and mutagenic potential (Kim et al., 2013). In fact, in the current European air quality legislation (European Fourth Air quality Daughter Directive 2004/107/EC), seven potentially harmful PAHs must, at least, be

monitored, but restrictions on PAH concentrations are currently solely limited to benzo[a]pyrene because of its recognized high toxicity (annual target value of 1 ng m$^{-3}$ in the PM10 particulate phase fraction (Pandey et al., 2011)). However, it is the conjunction of the PM intrinsic physico-chemical properties (e.g. nature of adsorbed PAH – Dachs and Eisenreich, 2000 – or

water affinity) with pressure, temperature, hygrometric variations or ageing processes in the atmosphere that ultimately condition phase partitioning (free vs bound fraction, Ravindra et al., 2006). Consequently, gas-phase PAHs which are relatively abundant – yet known to bear only weak carcinogenic or mutagenic effects (Nisbet and LaGoy, 1992) – can further react through gas-phase processes or heterogeneous gas-PM exchanges to produce noxious oxy- or nitro-PAHs, for instance (Atkinson and Arey, 1994; Bandowe et al., 2014). PAHs emitted in exhausts, in either the gas phase or the particulate phase,

must be analyzed and fully characterized at the same time to better understand their deposition mechanism or chemical transformation in the atmosphere and help ascertain their overall toxicity and impact on human health.

Several methods allowing the concomitant sampling of airborne PAHs in both the gas and particulate phases have been developed in recent decades (see e.g. the reviews by Pandey et al., 2011; Szulejko et al., 2014; Munyeza et al., 2019). The sampling protocol starts with the choice of a suitable sorbent material to either solely capture the vapor phase or solely retain

PM. The former sorbent material mostly consists of polyurethane foam, resins, or graphitized carbon black mesh, whereas the latter is made of glass fiber, quartz fiber, or Teflon. The sorbents are placed in series, i.e. one after the other in the exhaust line. The soluble organic fraction is then extracted off-line from the sorbent (filter and/or resin) for subsequent gas chromatography mass spectrometry (GC-MS) analyses (An et al., 2016; Elghawi et al., 2010; Sun et al., 2006). However, such solvent extraction methods exhibit recovery rates that are highly dependent upon the technique applied and the nature of PAHs

a priori present. Accordingly, the GC-MS method, which relies on solvent extraction methods and calibration standards, is a time-consuming technique which is inherently more sensitive towards compounds having the greatest solubility. To circumvent this limitation, solvent-free methods have been recently developed based on thermal desorption (e.g. Villanueva et al., 2018), microwave-assisted desorption, or solid-phase micro-extraction (Szulejko et al., 2014). However, because sampling substrates may differ for PM and gas trapping, and often necessitate extraction techniques before characterization

whose efficiencies are substrate-dependent, results obtained for the two phases may be difficult to compare and do not necessarily represent the whole PAH family making up either filter.

The CAST (Combustion Aerosol Standard) generator is often chosen to produce combustion-generated particles as it is easy to implement for systematic laboratory experiments with the fuel and oxidation air flows being easily modifiable, enabling the investigation of a variety of chemistries. A miniCAST soot generator operated with different parameters as a source of

combustion by-products can mimic some of the physico-chemical properties of e.g. aircraft emissions (Bescond et al., 2014; Marhaba et al., 2019; Moore et al., 2014). The observations derived from soot produced with this generator hence allows for potential real-world extrapolations, especially for combustion devices not equipped with after-treatment systems. Concomitantly sampling and characterizing the particulate and gas phases can thus be extremely useful when evaluating the impact of various sources (aircraft jet engines, wood combustion stoves, biomass burning) on the environment, as the

gas/particulate partitioning conditions the overall reactivity. The two-filter method presented here can therefore be utilized to

assess the efficiency of after-treatment systems, which are known to successfully remove the majority of particle-bound organic species from the surface, which results in increasing the elemental carbon (EC) to organic carbon (OC) contribution (Focsa et al., 2019). Time-of-Flight Aerosol Mass Spectrometry (ToF-AMS) has been used in the past by Ferge et al. (2006) and Mueller et al. (2015) to study PAH formation in a CAST generator at different oxidation flows. However, because only
the particle phase has been analyzed, no information about the gas phase composition can be derived in these experiments, which provide only an incomplete picture of the PAH family emitted in the exhausts.

In this work, we coupled a two-filter sampling method with a multi-technique analytical approach to fully characterize the particulate and gas phase compositions of combustion by-products. The two-filter collection method is intended to separate the particulate phase (Front Filter) from the gas phase (Back Filter) using fibrous filtration media (Quartz Fiber Filters – QFF).
Once collected, the filters are analyzed using a multi-technique approach encompassing two-step laser mass spectrometry (L2MS), secondary ion mass spectrometry (SIMS), and micro-Raman spectroscopy. The L2MS technique has been extensively developed in our group (at the University of Lille/PhLAM laboratory) over the last decade to specifically probe the chemical composition of combustion by-products (Delhaye et al., 2017; Faccinetto et al., 2011, 2015; Moldanová et al., 2009; Popovicheva et al., 2017). Its high sensitivity and selectivity towards specific classes of compounds owing to different
ionization schemes makes it an extremely valuable analytical tool that can be adapted to various samples. Using three different ionization wavelengths, it is possible to target various classes of compounds such as aromatic and aliphatic compounds. In addition, it is possible to reach a sub-fmol limit of detection for PAHs upon specific desorption and ionization conditions (Faccinetto et al., 2011, 2015). The laser desorption process along with its coupling with the subsequent ionization step have been optimized over the years (Faccinetto et al., 2008; Mihesan et al., 2006, 2008) and ensure a soft removal (with minimum
internal energy excess) of molecules adsorbed on the particle surface, while avoiding/limiting both their fragmentation and the in-depth damaging of the underlying carbon matrix (Faccinetto et al., 2015). L2MS spectra obtained in this work are additionally reinforced with the SIMS spectra of deposited miniCAST PM, with no sample preparation prior to the analyses since the particulate matter is preferentially trapped on the Front Filter. Subtle differences and similarities between Front and Back Filters are revealed using mass spectrometry measurements (L2MS and SIMS) and the recently developed advanced
statistical methodologies (Duca et al., 2019; Irimiea et al., 2018, 2019) based on principal component analysis (PCA).

## 2 Experimental methods

### 2.1 Sample collection

PM was sampled from the exhaust of a miniCAST generator (5201c) from Jing Ltd., as described previously in e.g. Yon et al. (2018). Briefly, the miniCAST contains a propane-nitrogen flame with operating conditions controlled by the flow rates of
propane, nitrogen, oxidation air ($Q^{air}$) and dilution air. The working points used in this study (and others in this series Bescond et al., 2014; Ouf et al., 2016; Yon et al., 2015) are detailed in Fig. 1. The main difference between these working points is the oxidation air flow and – for the SP4 point – nitrogen dilution, with an increasing oxidation air flow in the order $Q^{air}_{SP3} < Q^{air}_{SP2}$

$< Q^{air}_{SP4} < Q^{air}_{SP1}$. Note that the sole dilution system in our experimental setup is that of the miniCAST generator itself (dilution airflow 20 l min$^{-1}$, Fig. 1). The samples were deposited on quartz fiber filters (QFF, Pall Tissuquartz QAT-UP 2500) using a
specially designed sampling line (as illustrated in Fig. 1). A bypass line parallel to the sampling system has been added to ensure the miniCAST is maintained at atmospheric pressure. These QFF are typically used in soot collection, for example in studies of aircraft PM (Delhaye et al., 2017), and are also employed when deriving organic carbon to total carbon ratios (OC/TC) of deposited soot (Bescond et al., 2016; Yon et al., 2015). QFF are chosen because they proved to be highly efficient in capturing combustion emissions and they have a stable penetration curve among other filters when a range of physical
parameters are varying in the sampling line (Zíková et al., 2015). From a theoretical point of view, it is very difficult to predict the collection efficiency of QFF for particles within the nanometer size domain, as generated in our combustion conditions (e.g. 99–166 nm modal diameter, Bescond et al., 2016). Filter collection efficiency is directly related to inertial impaction, direct interception, Brownian diffusion and electrostatic forces (Brochot et al., 2019; Lindsley, 2016; Zíková et al., 2015). The resulting component of these forces is translated into a function that displays near 100 % collection efficiency for particles
smaller than 20 nm and larger than 300 nm. The minimum collection efficiency, which is also referred to as the most penetrating particle size (MPPS), is obtained for the 100–300 nm size range. However, some studies showed that the MPPS for QFF may peak around 60 nm and are possibly < 100 nm for other fibrous media (Brochot et al., 2019; Zíková et al., 2015). While these measured values are mostly influenced by both the flow velocity in the sampling line and the pressure drop at the surface of the filter, our flow conditions are close to those used in the work of Zíková et al. (2015). Consequently, we can
roughly estimate that the MPPS for the "Front Filter" is below 100 nm. Two filters were used for each sampling period: the "Front Filter" was a bare QFF placed in the exhaust line; the "Back Filter" was a QFF covered with a thin layer of black carbon (Pureblack 100 Carbon, Columbian Chemicals Company, specific surface area 80–150 m$^2$ g$^{-1}$) and placed 3.5 cm downstream of the Front Filter in the sampling line. Prior to sampling, the Back Filter was heated in an oven at 150°C for 16 hours to remove pre-adsorbed species. Back Filters thus produced were shown to yield no signal when analyzed by L2MS. Black carbon
has previously been used as a matrix upon which pure PAHs were adsorbed for mass spectrometric analysis of soot surrogates (Faccinetto et al., 2011, 2015). In the same studies, black carbon-covered filters were also used to sample the volatile fraction in flames. In the present study, the sampling line was designed to collect PM (including adsorbed species) on the Front Filter and to trap gas phase molecules from the remaining exhaust on the Back Filter. Note that particle build-up on the Front Filter could potentially increase its filtration efficiency and consequently trap PAHs that would instead pass through if the thickness
of the PM collected on the Front Filter were not as high. Alternatively, species originally adsorbed on the PM can also be desorbed during the sampling and be retained on the Back Filter, which would lead to an overestimation of the gas-phase fraction (Paolini et al., 2017). However, our results will show that if this is the case, only specific PAHs of intermediate volatility are impacted by this phenomenon. In addition, this effect would not affect our statistical analysis (i.e. the covariance between mass peaks ($m/z$)) as similar diffusion behaviors can be expected within SP1 and SP3 samples, which exhibit similar
soot porosity (e.g. the porosity of the soot material deposited on silicon wafers for SP1 and SP3 set points were calculated to be about 98.1 % and 97.4 %, respectively, Ikhenazene et al., 2019). We therefore expect, from a statistical standpoint, that for

each given $m/z$, the covariance will only negligibly be affected by diffusion. Sampling was performed for 20 minutes per working point. 'Reference' samples (Front and Back Filters) were collected by running the miniCAST generator for only two minutes under set point SP1 conditions. These samples represent pre-stabilization burner conditions. They were collected as a 'reference' to ensure that the samples were not impacted by this early combustion phase, and the loading on these 'reference' samples was much lower. After collection, samples were placed in watch glasses covered with Al foil, and stored at 4°C prior to analysis.

## 2.2 Two-step (desorption / ionization) laser mass spectrometry

Samples were analyzed using a two-step laser mass spectrometry (L2MS) technique built in-house (Mihesan et al., 2008). Briefly, the soot sample is introduced into the analysis chamber ($10^{-8}$ mbar) via a preparation chamber, where it is pre-cooled by a constant flow of liquid nitrogen in the sample holder to avoid the sublimation of the most volatile species. In the analysis chamber of the time-of-flight mass spectrometer (ToF-MS), the sample is irradiated at normal incidence by the beam of a frequency-doubled Nd:YAG laser (Continuum Minilite, $\lambda_d = 532$ nm, 4 ns pulsewidth) shaped using a circular aperture and a 10 cm focal length plano-convex $CaF_2$ lens to form a top hat beam, 0.8 mm-diameter spot on the sample surface. Such irradiation is known to induce the desorption of neutral species from soot without affecting the carbon matrix (Faccinetto et al., 2011). All samples were analyzed with same desorption conditions ($\lambda_d = 532$ nm, 400 µJ pulse$^{-1}$, 80 mJ cm$^{-2}$ i.e. 20 MW cm$^{-2}$).

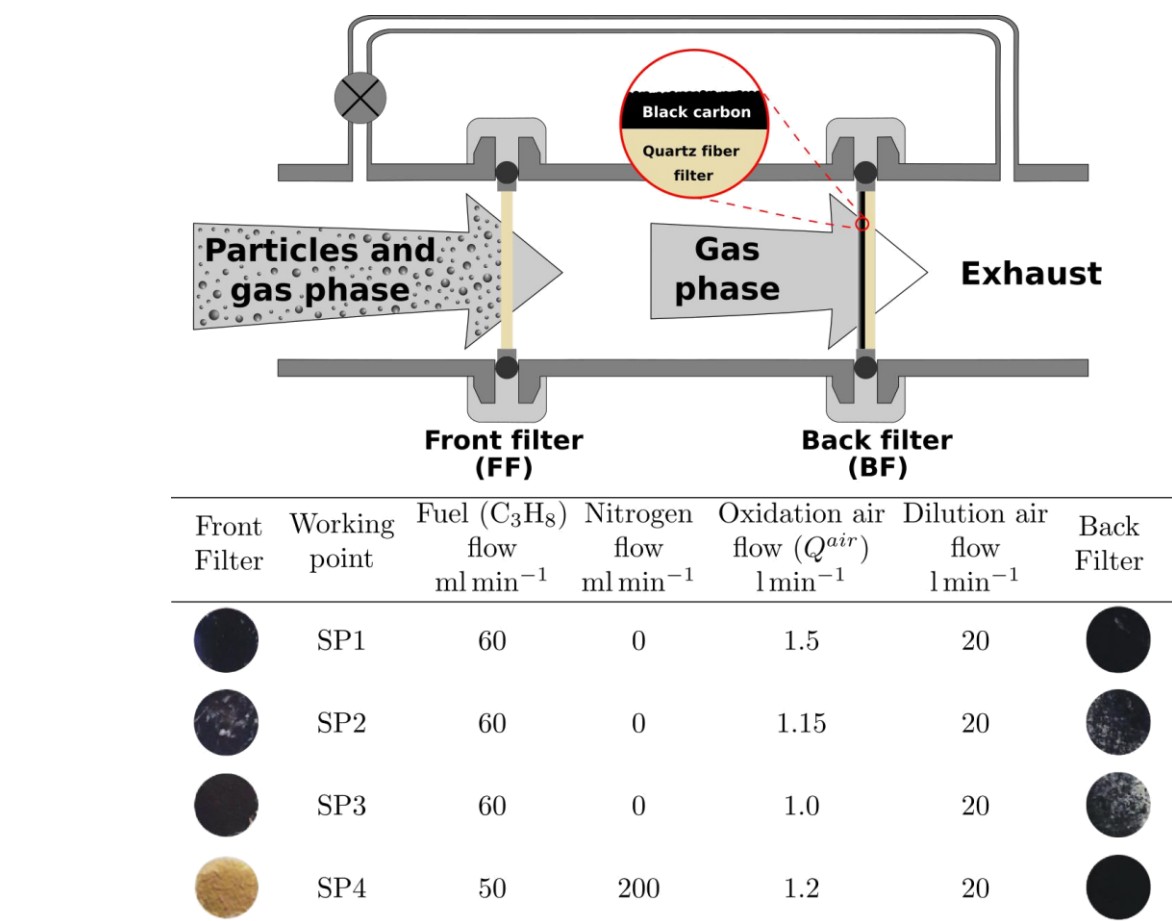

**Figure 1.** Schematic of the sampling line and photos of collected samples. The combustion parameters for the miniCAST burner are presented in the table.

The desorbed plume propagates normally from the sample surface in-between the extraction plates of the ToF-MS. The molecules from the desorbed plume are then ionized by either a resonant two-photon ionization R2PI (Haefliger and Zenobi, 1998; Mihesan et al., 2006; Zimmermann et al., 2001) process at $\lambda_i$ = 266 nm (4 ns-width pulsed UV laser, Continuum Powerlite, 1 mJ pulse[-1]) or a single photon ionization (SPI) process at $\lambda_i$ = 157 nm (5 ns-width pulsed VUV excimer laser, Coherent ExciStar XS 200) or $\lambda_i$ = 118 nm (in-house built coherent nanosecond source). The ninth harmonic of the Nd:YAG laser ($\lambda_i$ = 118.2 nm) was generated by tripling the 355 nm output of a Continuum Surelite pulsed laser in a Xe cell (Hilbig, 1982; Popovicheva et al., 2017). For the two SPI wavelengths, the setup was optimized to the maximum output (i.e. maximum electric potential for the 157 nm excimer laser, and maximum pumping energy for the 355 nm laser (34 mJ pulse[-1]) to maximize 118 nm conversion, at an estimated conversion efficiency of 0.01 % (Butcher, 1999). The time delay between desorption and ionization is set to 100 μs by a digital delay/pulse generator (Stanford DG535). Generated ions are then mass analyzed in a 1.72 m-long reflectron ToF-MS (RM Jordan) with a mass resolution of $m/\Delta m$ 1000. Ion detector signals are recorded using a digital oscilloscope (LeCroy Waverunner 9350AM) at a time resolution of 4 ns/point. Each spectrum corresponds to an average

of at least 200 desorption shots at different locations on the sample. A set of 3 spectra was obtained for each miniCAST sample (FF and BF) and each ionization scheme. Therefore, in total, 18 spectra were obtained for each miniCAST set point.

### 2.3 Secondary ion mass spectrometry

Time-of-Flight Secondary Ion Mass Spectrometry analysis was conducted with the TOF.SIMS[5] instrument from ION-TOF GmbH. Briefly, samples were introduced into the analysis chamber with a residual pressure of $10^{-8}$ mbar. The sample surface was bombarded by a 25 keV $Bi_3^+$ ion beam with a current of 0.3 pA in static mode. 180 s acquisition time and 25 random scans/acquisition were used for an analyzed area of 500 µm × 500 µm on the sample surfaces. Mass spectra were collected in both positive and negative polarities for at least three areas per sample. The mass resolution at *m/z* 29 is approximately 2700. For this analysis samples are not cooled down. Ion signals in SIMS mass spectra were identified and calibrated using SurfaceLab 6 software from ION-TOF GmbH. Positive spectra were calibrated with peaks $C^+$ (*m/z* 12.000), $CH_3^+$ (*m/z* 15.023), $C_7H_7^+$ (*m/z* 91.055), $C_{16}H_{10}^+$ (*m/z* 202.078), and $C_{19}H_{11}^+$ (*m/z* 239.086). Negative spectra were calibrated with peaks at $C^-$ (*m/z* 12.000), $CH^-$ (*m/z* 13.008), $O^-$ (*m/z* 15.995), $C_2^-$ (*m/z* 24.000), $C_4H^-$ (*m/z* 49.008), and $C_8H^-$ (*m/z* 97.008). A set of 5 spectra was obtained for each miniCAST sample (FF and BF) and each polarity. Therefore, in total, 20 spectra were acquired for each miniCAST set point.

### 2.4 Raman micro-spectroscopy

Raman analyses were performed with an Invia reflex spectrometer (Renishaw) equipped with an Olympus microscope (BXFM) (Chazallon et al., 2014). The spectra presented in this work were obtained by irradiation with a 514 nm laser with a nominal power of 150 mW. The laser power was reduced to avoid thermal effects at the sample surface. Using a lens with 20× magnification (N.A. 0.5), the laser was focused on the sample surface to a 3.0 µm diameter spot. The spectrometer was calibrated using the Stokes Raman signal of pure Si at 520 cm$^{-1}$. Raman spectra of spectral resolution 12 cm$^{-1}$ were collected at four different spots on each sample using integration times of 60 s with 10–20 scans accumulated per spectrum.

### 2.5 Multivariate data analysis: principal component analysis (PCA)

PCA is a technique used to highlight variation and patterns in a data set, and in this case was used to reveal the differences in chemical composition of the samples, and in particular between (i) Front and Back Filters and (ii) miniCAST set points. PCA is very convenient to outline the subtle differences between data sets, since it reduces the dimensionality of complex data while preserving most of the information. PCA was applied to each of the five datasets (3 L2MS ionization wavelengths and 2 SIMS polarities) following the procedure detailed in Popovicheva et al. (2017), Irimiea et al. (2018), and Duca et al. (2019). Further information can also be found in Sect. S1 of the supplementary material. Briefly, each mass spectrum was represented by the integrated areas of a selected number of mass peaks in the spectrum. The number of selected mass peaks was 66, 105, and 60 in L2MS mass spectra recorded at $\lambda_i$ = 266, 157, and 118 nm, respectively, and 138 and 70 in SIMS mass spectra recorded in

positive and negative polarity, respectively. PCA analyses were performed using a covariance matrix, i.e. each data set was organized into a matrix containing observations/samples (arranged in rows) and variables/peak integrated area (arranged in columns). Principal components (PCs) were constructed as linear combinations or mixtures of the initial variables (peak integrated areas). The physical meaning of all derived PCs can be inferred from the contribution of the various molecular species to the loadings, i.e. by determining the relative importance of each mass peak integrated area to the main variance in the data set. Scree plots and loadings for all L2MS PCA analyses discussed in this article can be found in the supplementary material (Fig. S1 and S2). It should be further noted that initial PCA tests included 'reference' samples (in both L2MS and SIMS PCAs). This preliminary step resulted in PC1 (the largest variance in the data set) being dominated by the variance between the 'reference' samples (Front and Back Filters) and all other samples, confirming that the two min pre-stabilization deposition does not influence the spectra of the various set points measured (see Sect. 2.1). After this confirmation step, the 'reference' samples were removed from the covariance matrix used to perform the PCA and therefore are not presented in the following sections.

## 3 Results and Discussion

### 3.1 L2MS analysis

#### 3.1.1 Mass spectra obtained by L2MS at individual ionization wavelengths

L2MS mass spectra of samples SP1, SP2, SP3, and SP4 (for both Front and Back Filters) produced at three different ionization wavelengths (266, 157, and 118 nm) are discussed in this section. Mass spectra obtained with 266 nm ionization wavelength are presented in Fig. 2, whereas results obtained for 157 and 118 nm are both presented in Fig. S3.

Upon 266 nm ionization, all mass spectra are dominated by signals attributed to aromatic species, and more specifically to PAHs (Fig. 2). An important advantage of L2MS is to generate, for the most part, fragment-free mass spectra while maintaining a high signal-to-noise ratio, due to the controlled desorption and ionization fluences (Faccinetto et al., 2011). On all mass spectra generated with 266 nm ionization wavelength, the lightest detected PAH is naphthalene ($C_{10}H_8^+$, $m/z$ 128).

On Front Filters ($\lambda_i = 266$ nm, Fig. 2), the heaviest detected mass varies from sample to sample: $SP1_{FF} - m/z$ 400, $SP2_{FF} - m/z$ 546, $SP3_{FF} - m/z$ 546, and $SP4_{FF} - m/z$ 522. The base peak is at $m/z$ 202 for $SP2_{FF}$, $SP3_{FF}$, and $SP4_{FF}$, and at $m/z$ 178 for $SP1_{FF}$. One can observe that the increase in oxidation air flow ($Q^{air}_{SP3} < Q^{air}_{SP2} < Q^{air}_{SP4} < Q^{air}_{SP1}$) results in a significant variation in the shape of the mass spectra. In the SP1 regime, most of the signal comes from three- and four-ring PAHs, while the heavier PAHs are less conspicuous. Regimes with the lowest oxidation air flow tend to produce more of the heavier PAHs, although the increase in contribution for each mass is not the same. For samples $SP2_{FF}$ and $SP4_{FF}$ most of the PAHs are concentrated in the mass range $m/z$ 178–350 ($C_{14}H_{10}^+$–$C_{28}H_{14}^+$), with comparable relative intensities. However, the SP3 regime has high peak intensities for $C_{14}H_{10}^+$ ($m/z$ 178), $C_{16}H_{10}^+$ ($m/z$ 202), $C_{18}H_{12}^+$ ($m/z$ 228), and $C_{20}H_{12}^+$ ($m/z$ 252), while the relative contributions of heavier PAHs remain comparable. Literature data converge towards the fact that the SP3 set point is distinct from the others in that i) the organic to total carbon ratio is higher (87 % versus ≤ 47 % for the other set points), and ii) the crystallites of the

particles produced in these conditions are significantly smaller and form a distinct disordered arrangement exhibiting many carbon edges (Bescond et al., 2016; Marhaba et al., 2019; Ouf et al., 2016; Yon et al., 2015). Such smaller crystallites suggest that SP3 may undergo nucleation and growth processes different from those of the other set points, subsequently leading to distinct chemical compositions (e.g. different isomeric distributions) of the PM. The relative ion signals observed between the Front and Back Filters hence depend upon the relative volatilities and the response of the chemical compounds present on the

samples to the 266 nm R2PI L2MS.

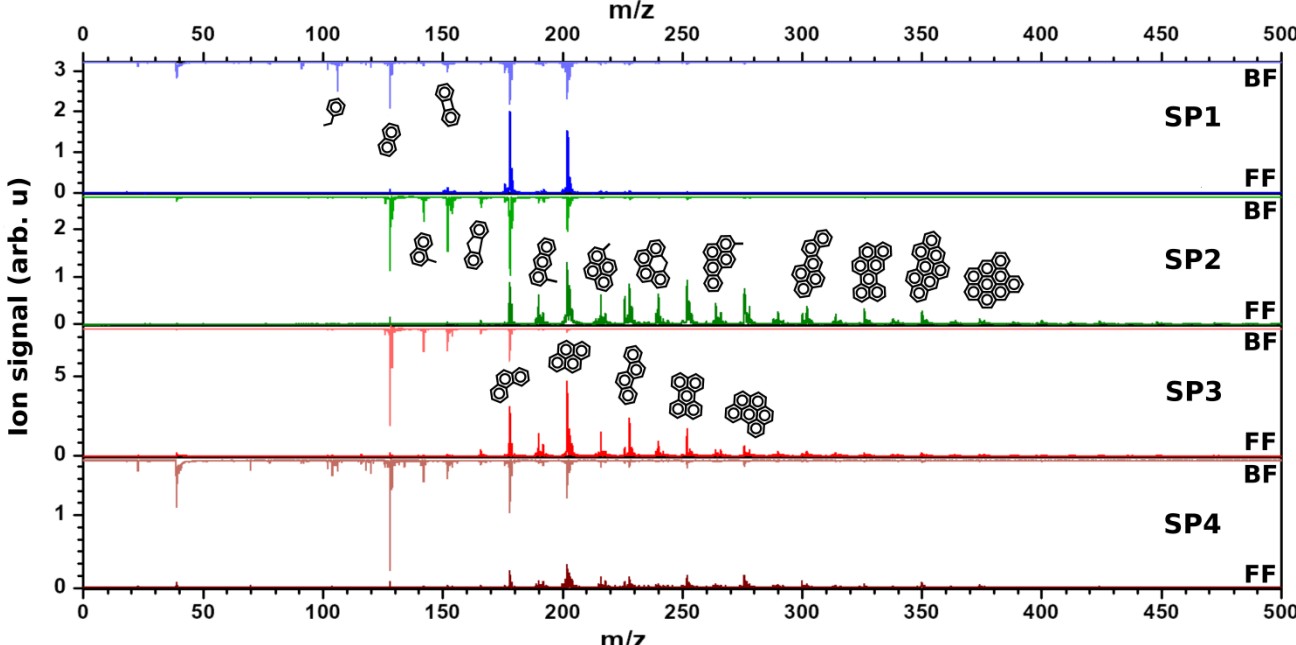

**Figure 2.** Comparison between mass spectra for SP1, SP2, SP3, and SP4 samples recorded with $\lambda_i$ = 266 nm for Front Filters (lower spectra) and Back Filters (upper spectra). Note that four different areas have been averaged to yield each of these spectra. Proposed structural formulae based on molecular formulae obtained from mass spectra are also shown.


On Back Filters ($\lambda_i$ = 266 nm, Fig. 2), the signal is mostly concentrated in a low mass region. Mass spectra of all Back Filters are dominated by $m/z$ 128 (naphthalene, $C_{10}H_8$). The spectrum of SP4$_{BF}$ shows more peaks of higher masses than what is observed on other samples. The first aromatic compound visible on all samples is benzene ($m/z$ 78), however its intensity is drastically reduced for both SP2$_{BF}$ and SP3$_{BF}$ compared to that of SP1$_{BF}$ or SP4$_{BF}$. The same trend is observed for aromatic

compounds lighter than $m/z$ 128. Oxidation air flows associated with set points SP4 and SP1 thus resulted in the formation of more of the smallest-sized aromatic species ($m/z$ 78–128).

The various miniCAST set points exhibit different PAH mass distributions on their Front and Back Filters, which likely relates to the different volatility properties of PAHs and probably affects their subsequent trapping on Front and Back Filters. Distinct volatility properties have been observed in the past on particles originating from wood combustion by Bari et al. (2010), who

classified the PAHs on the basis of their number of aromatic rings resulting in the detection of three different PAH categories. The authors classified the PAHs consisting of two aromatic rings as volatiles as they were mostly found in the gas phase, while those made of three and four rings were classified as semi-volatiles. PAHs comprising more than four rings were classified as non-volatile as they were observed in the PM in their study. Note that slightly different classes have also been defined elsewhere in the literature (An et al., 2016; Elghawi et al., 2010; Sun et al., 2006). In our study, we largely found compounds

consisting of one and two aromatic rings on Back Filters, while PAHs of $m/z$ 176–242 were found on both Back and Front Filters and those of $m/z \geq 252$ predominantly on Front Filters. Such PAH partitioning between Front and Back Filters is in line with the work of Bari et al. (2010). Similarly, we categorized the PAH distributions found on miniCAST samples into volatile, semi-volatile, and non-volatile fractions (Fig. 3), where the volatile fraction here encompasses aromatic species made of one to two aromatic rings ($m/z$ 78–166), the semi-volatile fraction comprises PAHs with a mass range of $m/z$ 176–242, and the

non-volatile fraction includes PAHs of $m/z \geq 252$. The boundaries of these intervals have been refined using the representation of Fig. 3 in which the "contrast function" defined as the $\frac{S_{FF}-S_{BF}}{S_{FF}+S_{BF}}$ ratio is represented for the 266 nm L2MS data, where $S_{FF}$ and $S_{BF}$ are the ion signals associated with a mass peak on the Front Filter and the Back Filter, respectively. This representation clearly underlines that small aromatic species are found solely on the Back Filters, whereas large PAHs are mostly on the Front Filters.

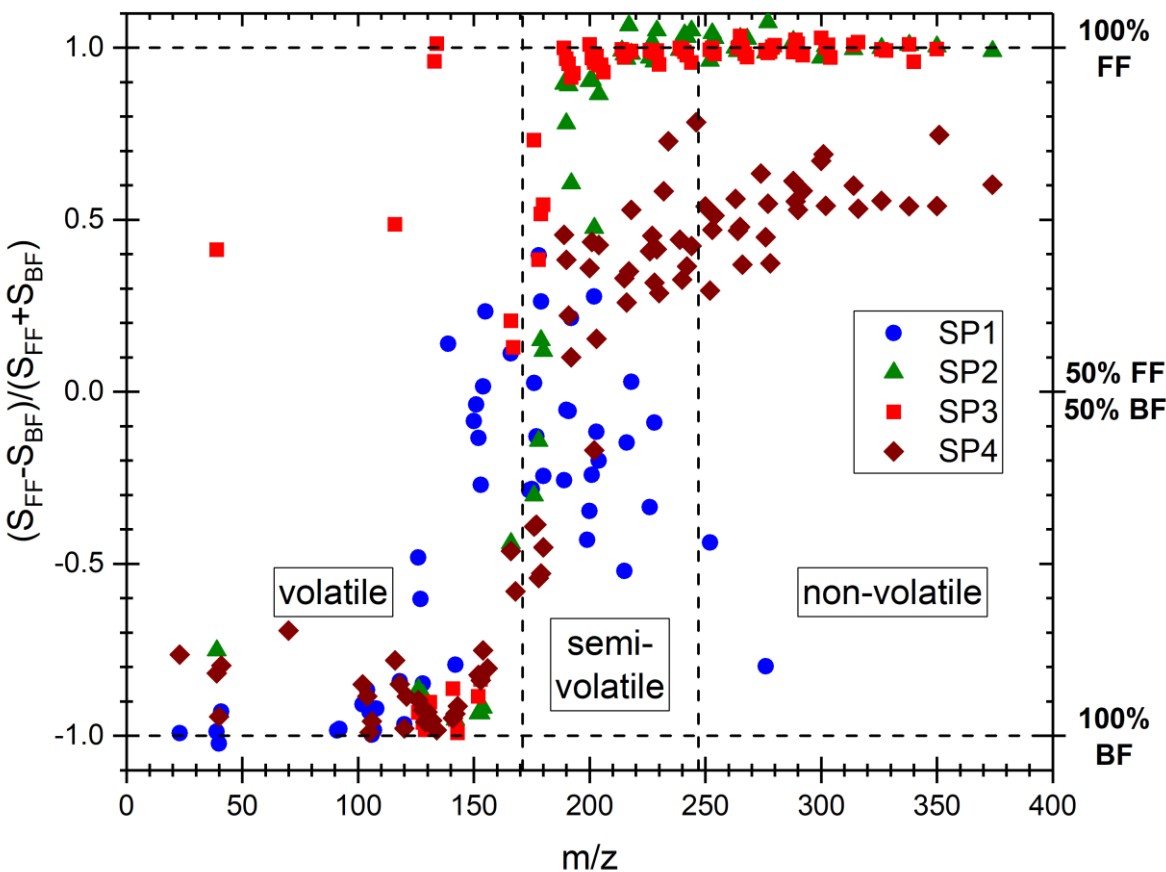

**Figure 3.** "Contrast plot" representing the variation in PAH signal detected with L2MS at $\lambda_i = 266$ nm for the four miniCAST set points. Values on the y-axis correspond to the partitioning of the species between the Front Filters and Back Filters: -1 indicates that the species are all found on the BF, +1 that they are all found on the FF, and 0 that they are equally partitioned on both filters.

300 The total PAH signal derived from L2MS measurements ($\lambda_i = 266$ nm) can be further refined according to the PAH mass range distribution present on each sample (Fig. 4) as previously defined in Fig. 3. Delhaye and coworkers showed that the total PAH signal in mass spectra obtained with $\lambda_i = 266$ nm can be indicative of the amount of organic carbon in aeronautical soot samples, because of the good agreement between total PAH mass signal and OC/TC values derived by a commonly used thermo-optical protocol (see Fig. 7 in Delhaye et al., 2017). In Fig. 4, the total PAH signal, corresponding to the sum of all peaks attributed

305 to PAHs in the 266 nm normalized mass spectra, is plotted against the oxidation air flow and is further compared to the OC/TC values given in Bescond et al. (2016) and Yon et al. (2015). According to these studies, SP1 has the lowest OC/TC ratio at 4.1 %, followed by SP4 (OC/TC 22.1 %), SP2 (OC/TC 46.8 %), and SP3 (OC/TC 87 %). Slightly different values for SP2 (OC/TC 58.3 %) and SP1 (OC/TC 16.2 %) are given by Yon et al. (2015), but the same overall trend is maintained. The same evolution

with oxidation air flow was evidenced for the PAH to soot ratio in Moore et al. (2014) using a different method (photoelectric

aerosol sensor). Figure 4 shows that the total PAH signal measured by mass spectrometry ($\lambda_i = 266$ nm) on Front Filters (orange bars) follows the same trend as the OC/TC ratios measured by the thermo-optical protocol for the same miniCAST set points (Bescond et al., 2016; Yon et al., 2015). Although the total PAH signal on Back Filters (blue bars, Fig. 3) also follows the trend observed on Front Filters for samples SP3$_{BF}$, SP2$_{BF}$, and SP4$_{BF}$ (decreasing PAH signal with increasing oxidation air flow), total PAH signal of SP1$_{BF}$ is high compared to that of SP1$_{FF}$. This is likely due to the nature of its PAH content for this

set point, which likely includes more volatile and semi-volatile aromatic species.

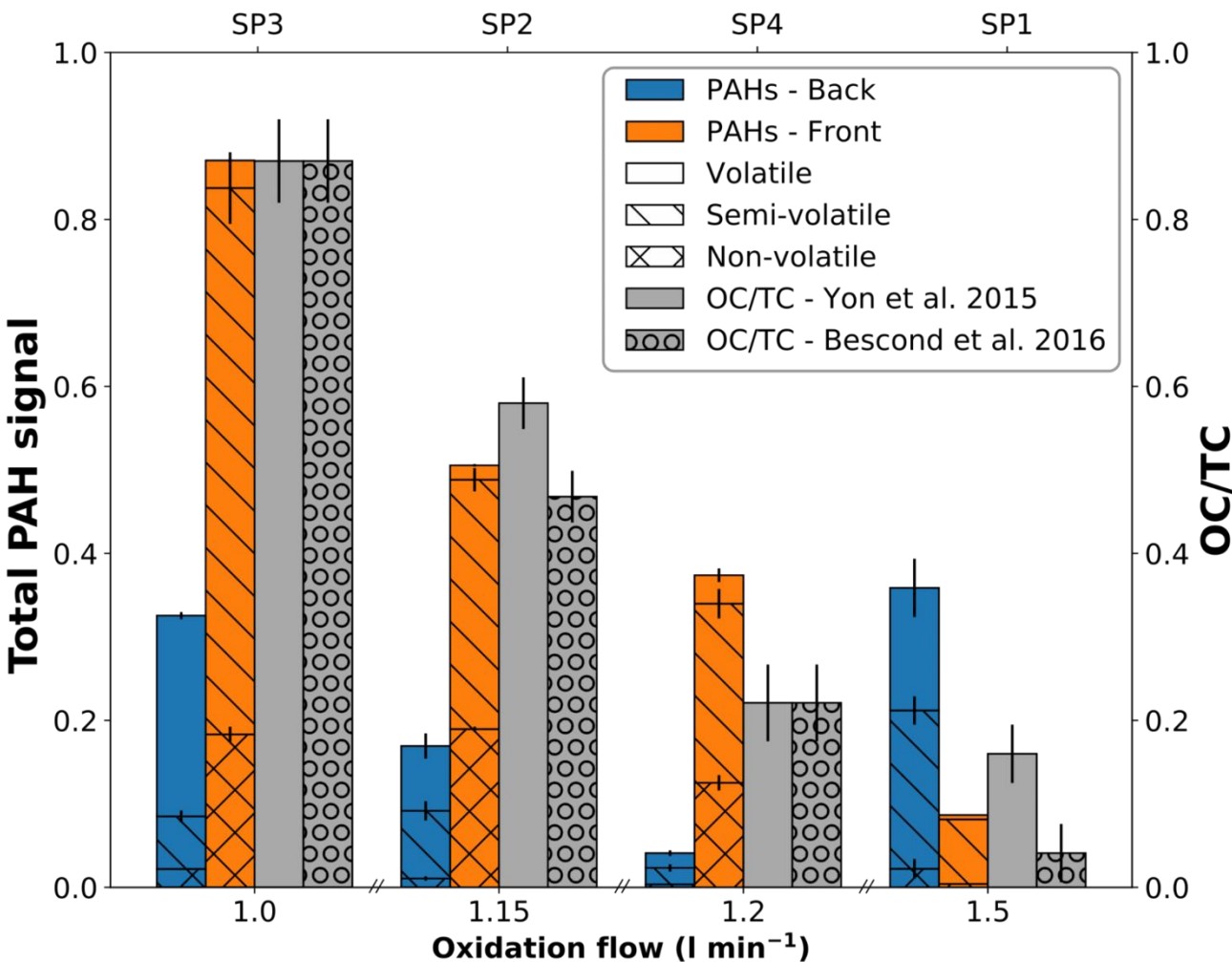

**Figure 4.** Variation in total PAH signal detected with L2MS at $\lambda_i = 266$ nm on Front (orange) and Back (blue) Filters plotted as a function of oxidation air flow. The PAH contribution is divided into adsorbed species (non-volatile, $m/z \geq 252$), semi-volatile ($m/z$ 176–242), and volatile ($m/z$ 78–166) fractions. OC/TC values reported in the literature are plotted in grey for comparison.

In order to access other classes of molecules, the miniCAST set points have been also analyzed using 157 and 118 nm ionization wavelengths. The majority of PAHs were also detected with SPI at 157 nm for both Front and Back Filters, albeit at a lower signal intensity (as can be seen by comparing the two sets of spectra in Fig. 2 and Fig. S3). The overall shape changes due to the different ionization efficiencies of PAHs from R2PI at 266 nm and SPI at 157 nm. At lower masses, additional peaks with prominent features at $m/z$ 28, 43, 55, 69 are present and are assigned to aliphatic fragment ions, which could result from multi-photon ionization processes. Analysis of the peak at $m/z$ 31 suggests the presence of heteroatoms in the fragments as it cannot be assigned to a $C_nH_m^+$ hydrocarbon formula. The series $m/z$ 91, 103, 115 corresponds to fragments ($C_7H_7^+$, $C_8H_7^+$, $C_9H_7^+$), which are attributed to alkylbenzene species (McLafferty and Tureček, 1993).

Mass spectra obtained with SPI at 118 nm (Fig. S3) show a high degree of fragmentation. In all cases, peaks at $m/z$ 23 and 39 are due to the presence of trace amounts of usual contaminants Na and K. PAHs were detected on all samples, but the signal intensity is low due to a high fragmentation rate caused by the excess of energy at $\lambda_i$ = 118 nm. Fragment ions at $m/z$ 50, 51, 52, 53, 63, and 65 suggest also the presence of aromatic compounds. Series of highly unsaturated aliphatic compounds ($C_{2n}H_2^+$ and $C_{2n}H_4^+$, $n$ = 2–5) are present. Fragments of alkyl compounds ($C_nH_{2n+1}^+$) were also found at $m/z$ 15, 29, 43, 57, 71, 85, and 99, with a relatively low intensity compared to fragments of aromatic compounds. Another distinctive series – attributed to $C_nH_{2n}^+$ fragments – was found at $m/z$ 28, 42, 56, and 70. These fragments may result from a McLafferty rearrangement involving alkene chains (McLafferty and Tureček, 1993).

In conclusion, our L2MS results for the three ionization wavelengths converge to show that heavy PAHs ($m/z \geq 252$) are largely found on the Front Filters, whereas the volatile aromatic species (1–2 rings) are solely detected on the Back Filters. This detailed mass spectrometry analysis coupled with our two-filter collection system shows very effective partitioning of the PM/gas phases on FF and BF and a clear dependence on the volatility of the molecules. We find that the total PAH content varies with the oxidative air flow, as shown in previous studies. Finally, this two-filter system allows us to evidence subtle differences in the chemical composition of the various miniCAST set points.

### 3.1.2 Principal component analysis of L2MS spectra

In order to better discriminate the chemical composition of the various samples, particularly (i) the Front and Back Filters and (ii) the miniCAST set points, principal component analysis (PCA) was applied to mass spectra recorded for all three individual ionization wavelengths. A full description of this statistical method is provided in Sect. S1. Here, the covariance matrix was built from the integrated areas of all the detected peaks with a signal-to-noise ratio SNR > 3. The physical meaning of all derived principal components can be inferred from the contribution of the various molecular species to the loadings (see Sect. S1 and Fig. 5b and S2). By identifying the molecular families contributing to this variance, we can interpret the PCA score plots (Fig. 5) and grasp the nature of the subtle chemical differences between the samples.

The loading and scree plots corresponding to the L2MS data generated with the 266 nm ionization wavelength are presented in Fig. 5b and S1a, respectively. They show that PC1 expresses the largest variance (58.86 %) in the dataset and differentiates samples having a large amount of high-mass PAHs (positive contribution: $m/z \geq 189$) from those containing more of low-mass aromatic species (negative contribution: up to three aromatic rings), especially naphthalene ($m/z$ 128). PC2 (19.30 %) denotes the relative contribution between high-mass PAHs (positive contribution: $m/z \geq 216$) bearing four and more aromatic rings, and aromatic compounds containing up to $m/z$ 202, especially m/z 178 and 202, and benzene and its alkyl-derivatives. The dataset, in terms of PC1 and PC2, is illustrated in a score plot in Fig. 5a. According to PC1, the largest separation appears between sample groups SP2–4$_{FF}$ and SP1–4$_{BF}$. It can be attributed to the higher fraction of high-mass PAHs ($m/z \geq 189$) relative to smaller aromatic species in SP2–4$_{FF}$. The first conclusion is that our samples are mainly separated regarding their chemical composition (non-volatile and semi-volatile fractions vs volatile fraction) because of the two-filter collection system rather than the miniCAST operating conditions. However, a refined observation in the PCs can help to interpret composition variations between the different set points. For Back Filters, the PC1 score decreases along with the oxidation air flow indicating a greater contribution of small aromatic species, especially naphthalene, for lower oxidation air flows. Data points for Front Filters generally display a positive PC2 component except for SP1$_{FF}$, a phenomenon possibly explained by the very small fraction of non-volatile PAHs produced in this regime relative to $m/z$ 178 and 202. The almost constant score of PC2 for SP2–4$_{FF}$, which is in contrast with the very different scores for the Back Filters (Fig. 5a), and the high contribution of semi-volatile species to the PC2 loadings (Fig. 5b) highlight that the ratios of semi-volatile compounds vary between the Front and Back Filters for the different set points. This observation suggests that the partitioning between the Front and Back Filters is not only driven by thermodynamic conditions (volatility) but also by the nature of the soot matrix produced at the different set points. Note that details about the PCA applied to the 157 nm and 118 nm L2MS data can be found in the supplementary material (Sect. S1). The statistical approach developed in this section confirms from a quantitative standpoint the descriptive results obtained in Sect. 3.1.1.

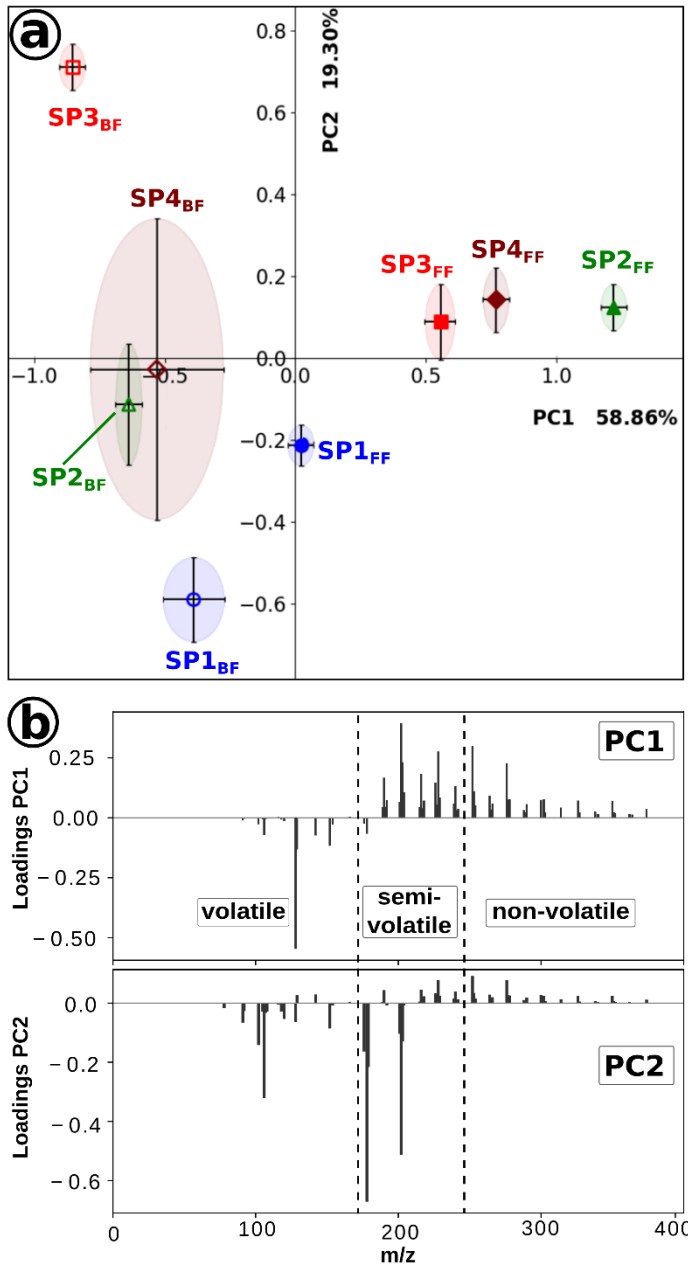

**Figure 5.** Score (a) and loading (b) plots of PC2 and PC1 derived from principal component analysis of the L2MS data obtained with 266 nm ionization.

### 3.2 SIMS analysis

#### 3.2.1 Mass spectra obtained by SIMS

SIMS measurements are complementary to L2MS analysis as they can provide insights into the compounds that preferentially produce negative ions. For the sake of comparison with L2MS results, SIMS measurements were first obtained in positive mode. Positive spectra of SP1, SP2, SP3, and SP4 samples are presented in Fig. 6 ($m/z$ 150–500 range). All SIMS mass spectra feature significant fragmentation, which is intrinsic to the technique and caused by the energetic primary ion beam. To start with Front Filter samples, and similarly to what has been observed in L2MS, the PAH distribution across samples varies with

set points, the highest detected mass on $SP1_{FF}$, $SP2_{FF}$, $SP3_{FF}$, and $SP4_{FF}$ being $m/z$ 452, $m/z$ 871, $m/z$ 825, and $m/z$ 908, respectively. $SP1_{FF}$ shows a shorter high-mass PAH "tail" compared to other Front Filter samples. The high-mass PAH region observed in SIMS is in good agreement with that of L2MS, whereas the significant fragmentation observed in SIMS seems to impair the low-mass region (ca. $m/z < 228$) more conspicuously and as a result makes SIMS and L2MS spectra look more distinct in this region. It is worth recalling that SIMS measurements are performed at room temperature, which contrasts with

L2MS measurements that involve nitrogen cooling. This may potentially result in SIMS analyses providing an incomplete picture for some specific low-mass PAHs. The base peak of $SP2_{FF}$, $SP3_{FF}$, and $SP4_{FF}$ samples is located at $m/z$ 239 ($C_{19}H_{11}^{+}$), whereas $SP1_{FF}$ exhibits its highest intensity peak at $m/z$ 202 ($C_{16}H_{10}^{+}$). The absolute intensity of the overall signal is the highest for $SP3_{FF}$, followed by $SP2_{FF}$, $SP4_{FF}$, and $SP1_{FF}$. Likewise, the total PAH contribution in each sample (i.e. summed areas of all peaks attributed to PAHs in positive mode) decreases with the oxidation air flow rate for Front Filters, as shown in Fig. 7,

which indicates that the general trend previously shown with L2MS is also observed with SIMS. Consequently, the total PAH signal derived from SIMS measurement is also in line with OC/TC measurements obtained from thermo-optical methods (Bescond et al., 2016; Yon et al., 2015) for all Front Filter samples (Fig. 7). Here, non-volatile species are predominant on $SP3_{FF}$, $SP2_{FF}$, and $SP4_{FF}$, while semi-volatiles constitute the main class of compounds observed on $SP1_{FF}$. The prevalent class of chemical compounds in L2MS or SIMS mass spectra (Fig. 4 vs Fig. 7) is linked to the ionization process ($\lambda_i = 266$ nm vs

$Bi_3^{+}$ ion beam). Semi-volatile compounds ($m/z$ 178–228) are prevalent in spectra generated with 266 nm ionization wavelength, whereas high-mass aromatic compounds ($m/z$ 250–500) dominate SIMS mass spectra. This suggests that miniCAST samples contain PAHs with a high stability in the high-mass range (i.e. stabilomers, Stein and Fahr, 1985), which are eventually less prone to fragmentation in SIMS with respect to other semi-volatile compounds. Consequently, a smaller relative fraction of molecules is fragmented in L2MS which leads to a more reliable PAH content determination in the semi-volatile mass range

from their L2MS mass spectra compared to SIMS.

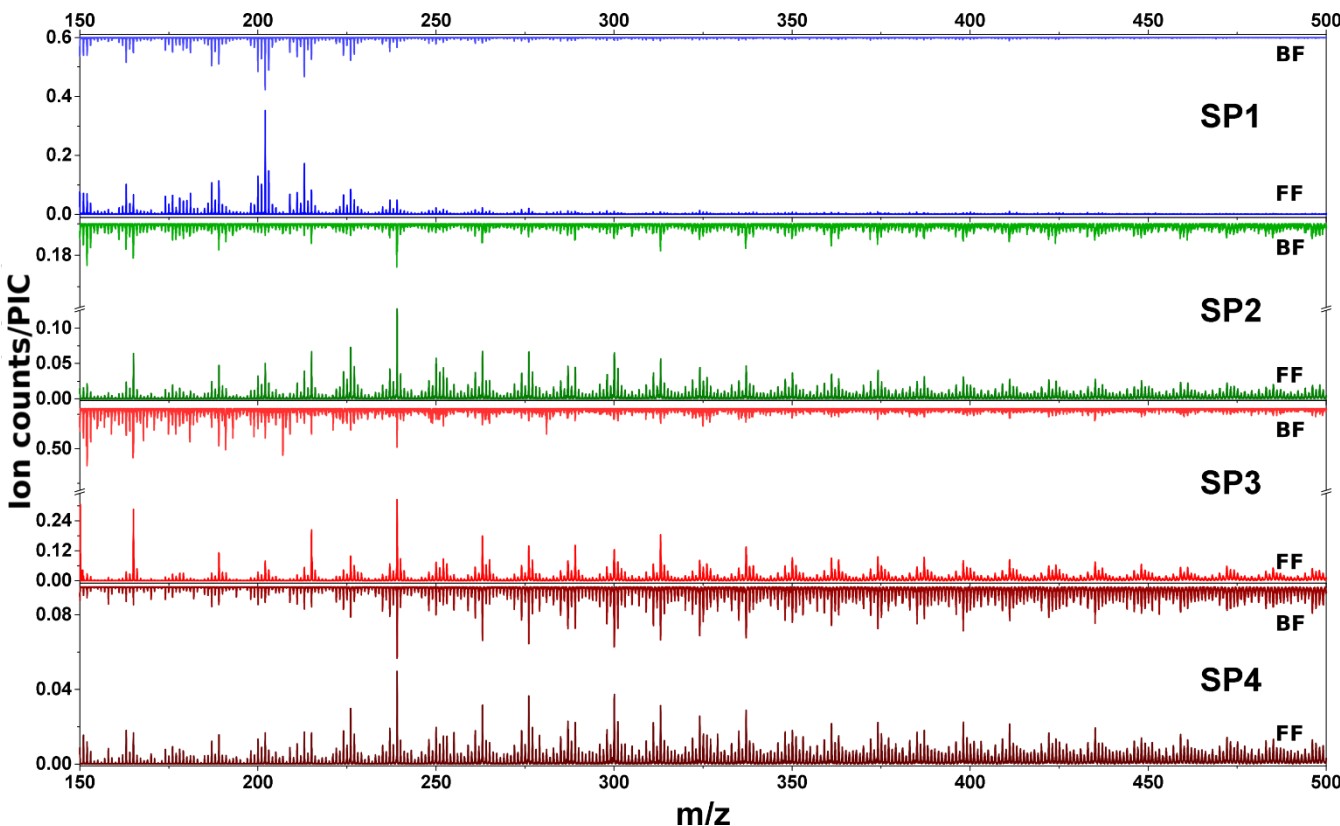

**Figure 6.** ToF-SIMS mass spectra of samples SP1, SP2, SP3, and SP4 obtained in positive polarity for Front Filters (lower spectra) and Back Filters (upper spectra). For visualization purposes, we focus on the *m/z* 150–500 range. Note that for SP2 and SP3 samples Front and Back filters have different scales.

As to the PAH distribution observed on Back Filters, it is distinct from that of Front Filters in that the highest mass detected is *m/z* 411 on SP1$_{BF}$, *m/z* 570 on SP2$_{BF}$, *m/z* 448 on SP3$_{BF}$, and *m/z* 793 on SP4$_{BF}$ (i.e. the *m/z* distribution of Back Filters is less spread out towards high masses). The total PAH signal shows now a different behavior to that observed on Front Filters (Fig. 7), where SP4$_{BF}$ exhibits the highest PAH signal, followed by SP2$_{BF}$, SP1$_{BF}$, and SP3$_{BF}$. Additionally, the PAH signal of SP4$_{BF}$ determined by SIMS is higher than the one of its corresponding Front Filter, which is in contradiction to what has been derived from L2MS ($\lambda_i$ = 266 nm) spectra, where the Front Filter showed a much higher PAH signal. This behavior may originate from the nature of deposited PAHs on Front and Back Filters, which may have different volatility and stability properties and hence will react differently to the energetic $Bi_3^+$ ion beam used in SIMS analysis.

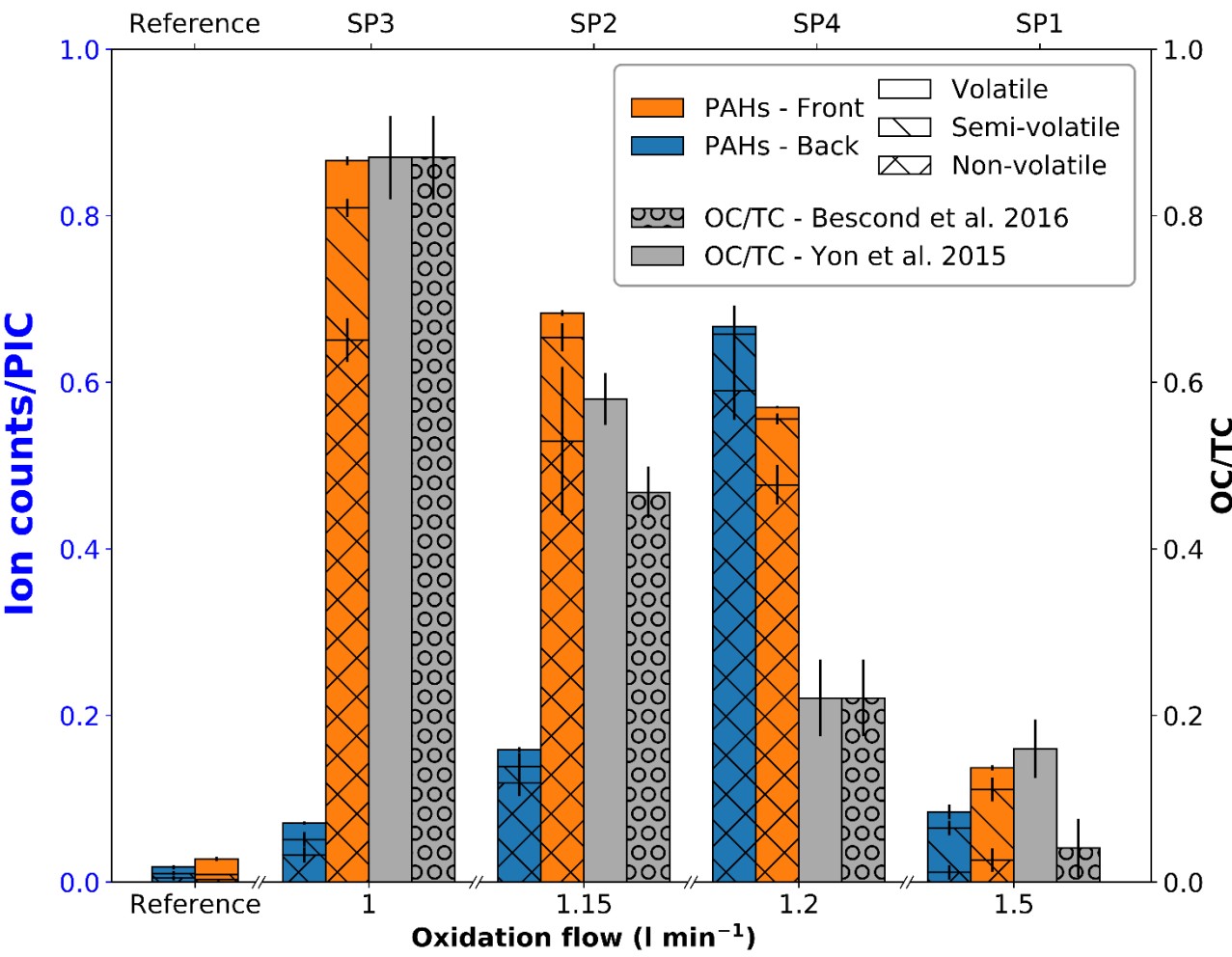

**Figure 7.** Total PAH signal detected with SIMS on Front (orange) and Back (blue) Filters plotted as a function of oxidation air flow, along with OC/TC values reported in the literature for the same miniCAST set points. All values are normalized to the partial ion count (PIC) corresponding to the signal of all selected peaks.


Negative polarity mass spectra obtained for Front and Back Filters are presented in Fig. S4. $H^-$ and $C_2^-$ fragment ions have the highest abundance in mass spectra of $SP1_{FF}$ and $SP1_{BF}$, whereas $SP2_{FF}$, $SP3_{FF}$, $SP4_{FF}$, and $SP4_{BF}$ are dominated by $H^-$ and $C_2H^-$. Similarly to that of the "reference" sample, spectra of $SP2_{BF}$ and $SP3_{BF}$ are dominated by $H^-$ and $OH^-$ ions. The $C_nH^-$ series is observed in mass spectra of SP2, SP3, and SP4 samples (Front and Back Filters), with the intensity decreasing with

the mass (for $n \geq 2$). To further understand this behavior, the relative abundances of the $C_n^-$ and $C_nH^-$ fragments are plotted as a function of the oxidation air flow in Fig. 8. $C_n^-$ fragments are commonly considered as markers of the EC content (Duca et al., 2019; Pagels et al., 2013; Popovicheva et al., 2017). For instance, Pagels et al. (2013) used the sum of $C_n^-$ ($n = 2$–4)

fragment signals as a marker of EC in aerosol time-of-flight mass spectrometer (ATOFMS) measurements for wood stove PM emissions and the same marker ions have been used to discriminate diesel from biodiesel PM emissions by Popovicheva et al.

(2017). $C_nH^-$ fragments are commonly associated with OC contents (Ewinger et al., 1991; Le Roy et al., 2015), but it is worth noticing that both series have been detected in mass spectra of pure PAHs (Bentz et al., 1995; Le Roy et al., 2015). Here, the relative proportion of $C_n^-$ fragments increases with the initial oxidation air flow conditions on Front Filter samples and exhibits a similar contribution across Front and Back Filter samples. It should be noted that the black carbon, pre-deposited on Back Filters, also contributes to the $C_n^-$ signal. However, for all miniCAST samples, the total $C_n^-$ contribution is higher than that of

the "reference" sample, suggesting that some proportion of $C_n^-$ on Back Filters originates from the deposited material. On the other hand, the $C_nH^-$ ion series shows a distribution across Front and Back samples akin to that of the total PAH signal (i.e. the total PAH signal in positive polarity, see Fig. 7). This positive correlation indicates a possible polyaromatic origin of the $C_nH^-$ fragments. In order to better delineate the contributions of EC, PAHs, and other components to the $C_n^-$ carbon cluster series, Pearson correlation coefficients between all $C_n^-$ and $C_nH^-$ have been evaluated for the complete set of negative SIMS

spectra. $C_n^-$ ($n$ = 1–4) ions display high positive correlation ($r \geq 0.60$) with a maximal value for $C_3^-$ and $C_4^-$ ($r = 0.91$). In contrast, this group of peaks is not correlated with the $C_5^-$ fragment ion and is anticorrelated with all heavier carbon cluster ions ($n \geq 6$). This first analysis shows that at least two components contribute to the $C_n^-$ signal. Furthermore, positive correlations are also found between the $C_nH^-$ ($n$ = 3–12) fragments, but also between these $C_nH^-$ ions and the $C_n^-$ ($n$ = 5–12) fragments. For better visibility, specific subsets of $C_n^-$ fragment ions ($n$ = 1–4 and $n$ = 5–12) are plotted separately in the

lower panels of Fig. 8. The lower left panel shows that the ions with carbon numbers $n$ = 1–4 primarily contribute to the total $C_n^-$ signal, whereas the lower right panel highlights the correlation between subset ions with carbon numbers $n$ = 5–12 and $C_nH^-$ fragments. In addition, the similar distribution across Front and Back Filters between $C_nH^-$ fragment ions and total PAH signal which is a proxy for OC supports the fact that $C_nH^-$ fragments can be considered as a marker for OC in our soot samples. Therefore, while a predominant fraction of $C_n^-$ fragments ($n$ = 1–4) are markers for EC, a non-negligible part ($C_n^-$

with $n \geq 5$) also originates from the organic fraction present on our samples.

SIMS results confirm L2MS measurements regarding the organic carbon and more specifically the PAH contents and mass distributions for the various miniCAST set points. In contrast to L2MS, specific SIMS fragmentation patterns provide additional information about the presence of elemental carbon and outline the distinct elemental carbon vs organic carbon contents for the different miniCAST set points.

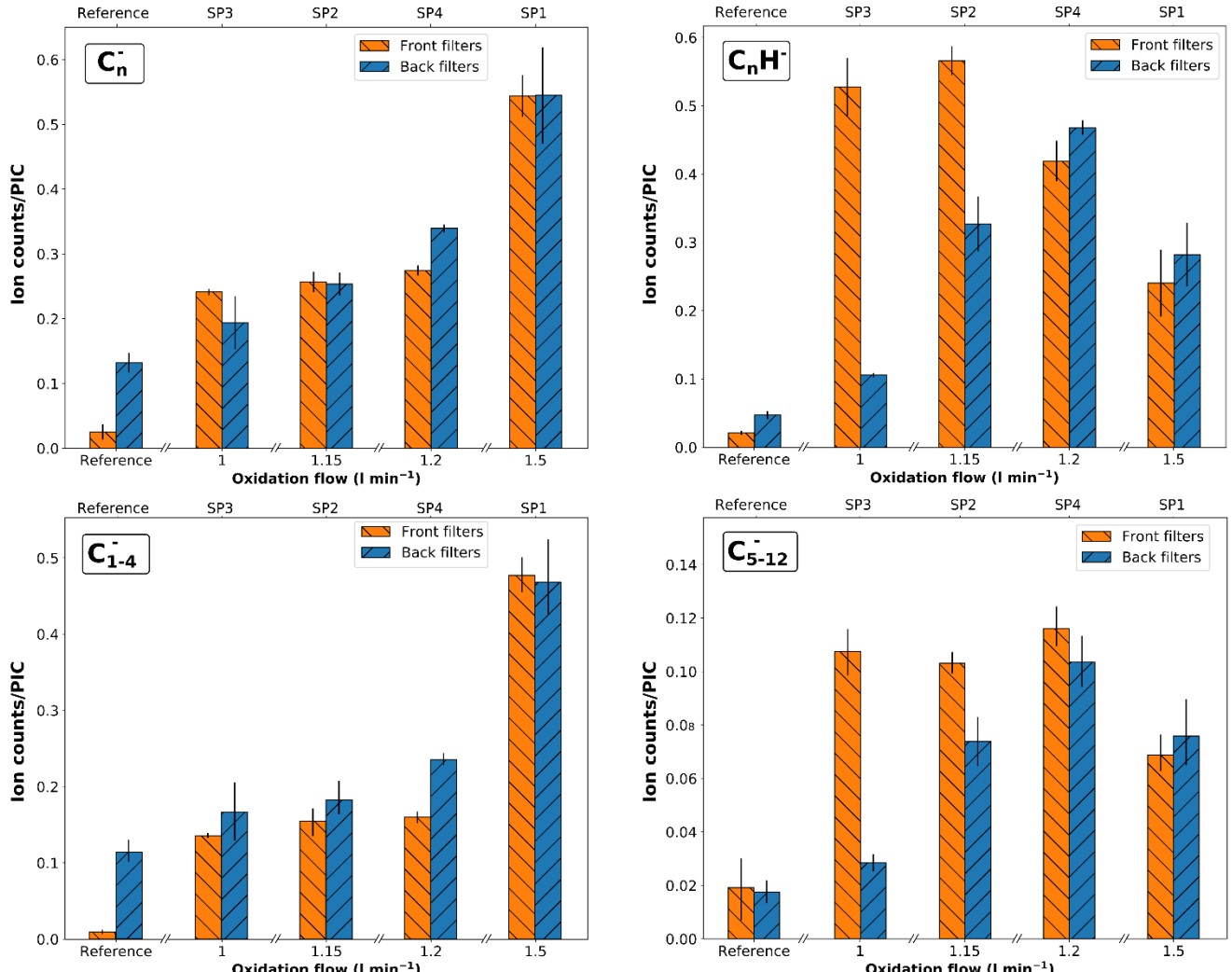

**Figure 8.** Variation of the signal of various markers, as derived from SIMS spectra. The panels represent the total peak areas of the following families: (upper left) $C_n^-$, (upper right) $C_nH^-$, (lower left) $C_{1-4}^-$, and (lower right) $C_{5-12}^-$. All values are normalized to the partial ion count (PIC) corresponding to the signal of all selected peaks.

### 3.2.2 Principal component analysis of SIMS spectra

PCA was applied to the positive mode SIMS spectra. All hydrocarbon fragments and the most representative peaks for PAHs were chosen for the analysis (see Sect. S3). The PCA score plot for the first two components (PC1 and PC2, responsible for 92 % of the variance) is presented in Fig. 9a, and their corresponding loadings in Fig. 9c. PC1 represents 73 % of the variance and is associated with small fragment ions with $m/z < 160$ (e.g. $C_nH_3^+$ with $n = 1$–3, $C_nH_m^+$ with $m > n$, $C_7H_7^+$, positive coefficients), and with polyaromatic species with $m/z \geq 165$ (negative coefficients). All Back Filter samples (containing gas-

phase PAHs) except SP4$_{BF}$ have positive PC1 scores, whereas all Front Filters but SP1$_{FF}$ exhibit negative PC1 scores due to their high PAH contents. From this result, it can be determined that SP3$_{FF}$ has the highest relative PAH content. Among Front Filters, SP1$_{FF}$ has the smallest contribution from PAHs. SP3$_{BF}$ has the highest contribution from fragments associated with both aliphatic and aromatic species. The negative contribution of PC2 (19 % of the variance) is associated with the hydrogen-poor fragments ($C_nH_m^+$ with $m < n$) and volatile and semi-volatile PAHs ($m/z$ 128–228) with the exception of $m/z$ 165. The positive coefficients of PC2 are associated with aliphatic fragments (e.g. $C_nH_m^+$ with $m > n$) and large PAHs ($m/z \geq 239$). As most of the variance was contained in only two principal components (92 %), there are only two available criteria for differentiating between samples: PC1, corresponding to the amount of PAHs relative to fragments, and PC2, depending partially on the hydrogen content of fragments originating from aromatic and aliphatic species. SP1$_{FF}$ and SP1$_{BF}$ are located in almost the same position due to their positive scores in PC1 and negative scores in PC2, which corresponds to their limited high-mass PAH content and high fragment content on the one hand, and small PAHs and hydrogen-rich fragments on the other hand.

PCA was also applied to the negative mode SIMS spectra for selected mass peaks, including carbon clusters $C_n^-$, $C_nH^-$, and some oxygenated and nitrogenated fragment ions. The first two components were determined to represent more than 85 % of the variance. The score plot of PC1 vs PC2 is presented in Fig. 9b, along with their corresponding loadings (Fig. 9d). The first component, which accounts for over 67 % of the variance, separates the samples containing low-mass carbon clusters $C_n^-$ (with $n \leq 3$), nitrogen and oxygen bearing compounds (e.g. $CN^-$, $C_3NH^-$, $CH_3O^-$) (positive PC1) from the samples containing species with a higher mass ($C_nH_{0-2}^-$ with $n \geq 4$, oxygenated, and nitrogenated fragments – negative contribution). The opposite contributions to PC1 of small carbon clusters in comparison to larger ones, with a transition size of $n = 3$–4, strengthen their dichotomous origin as already discussed in Sect. 3.2.1. In the light of the score plots, SP1$_{FF}$ and SP1$_{BF}$ samples are characterized by low surface coverages where small carbon clusters are associated with the soot matrix and the black carbon respectively, whereas the other samples feature PAH-rich surfaces. PC2, accounting for 18 % of the variance, separates data points based on the contribution from hydrocarbon compounds ($C_nH_m^-$, negative PC2) and oxygen/nitrogen bearing fragments (positive contribution). PC2 distinguishes SP3$_{BF}$, and SP2$_{BF}$ to a lesser extent, by their coverage in oxygen and nitrogen-containing species.

To sum up, PCA on SIMS results confirms the existence of various families of carbon clusters on the PM that can be associated either with the soot matrix or the surface PAH coating.

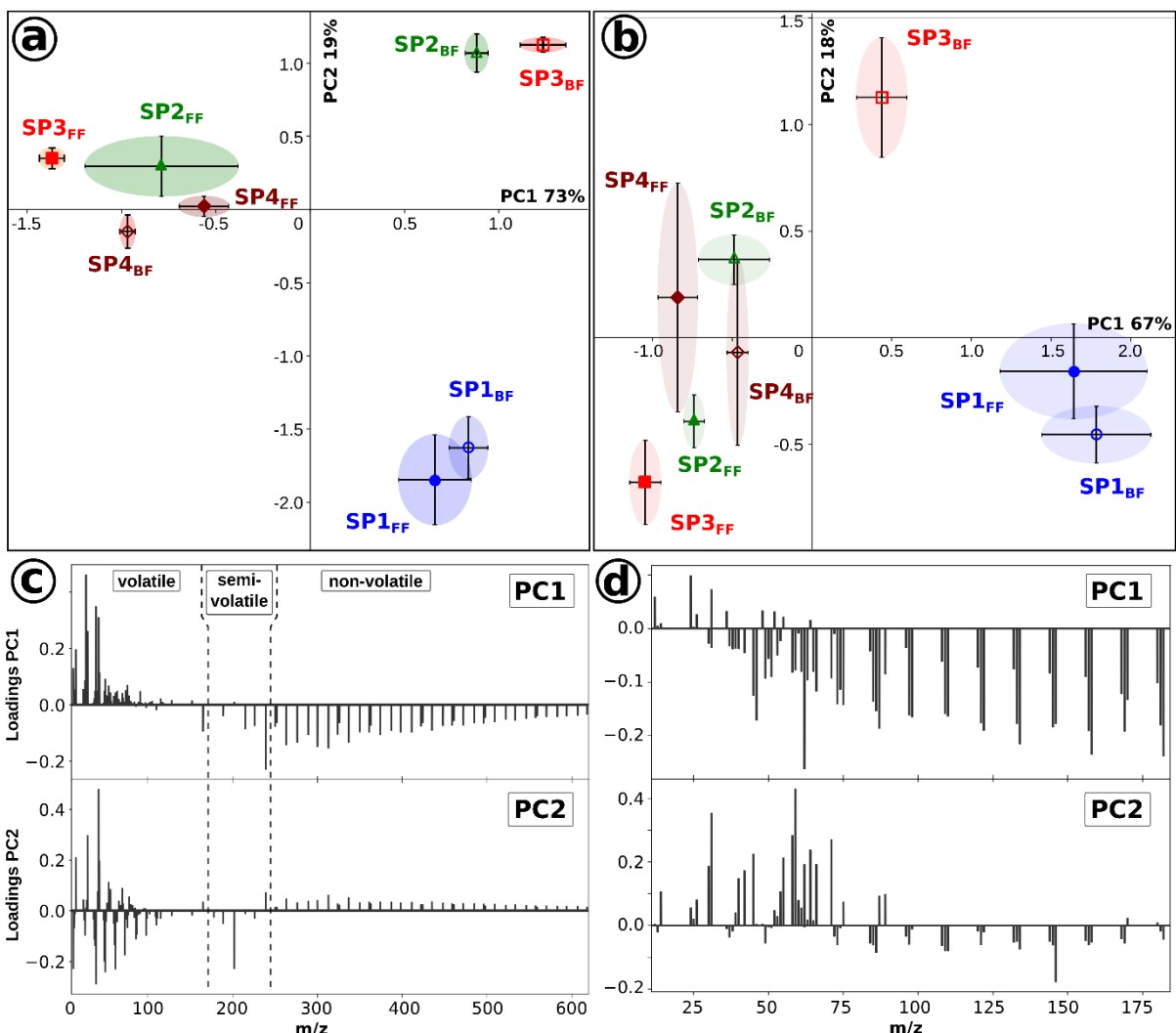

**Figure 9.** Score plots of PC1 and PC2 derived from positive (a) and negative (b) polarity SIMS mass spectra of miniCAST soot samples SP1, SP2, SP3, and SP4 (Front and Back Filters). Loadings corresponding to the contribution of different species to PC1 and PC2 derived from positive (c) and negative (d) polarity SIMS mass spectra.

### 3.3 Raman micro-spectroscopy analysis

The two-filter system provides a unique opportunity to perform Raman spectroscopy on either the gas phase trapped on the Back Filter or the PM collected on the Front Filter. Raman spectra measured for each sample are presented in Fig. 10. All spectra for PM deposited on Front Filters are in very good agreement with those already measured for the same miniCAST set points (e.g. Ess et al., 2016), while Back Filter spectra are dominated by the absorption of the pre-deposited black carbon. Soot

500

505

particles often exhibit distinct Raman signatures that can be used to distinguish samples mostly by their hybridization and nanostructure (e.g. stacking properties) compared to that obtained for a perfect graphite crystal, i.e. a crystal made of $sp^2$-hybridized carbons and graphene sheets stacked with their surfaces parallel and slightly offset. When samples differ from perfect crystalline graphite, defects appear and can take the form of stacking disorder (such as in turbostratic arrangements where tortuosity reduces the stacking order), edge sites, missing atoms in the graphite lattice or even altered local or semi-long range arrangements of carbon atoms (Parent et al., 2016).

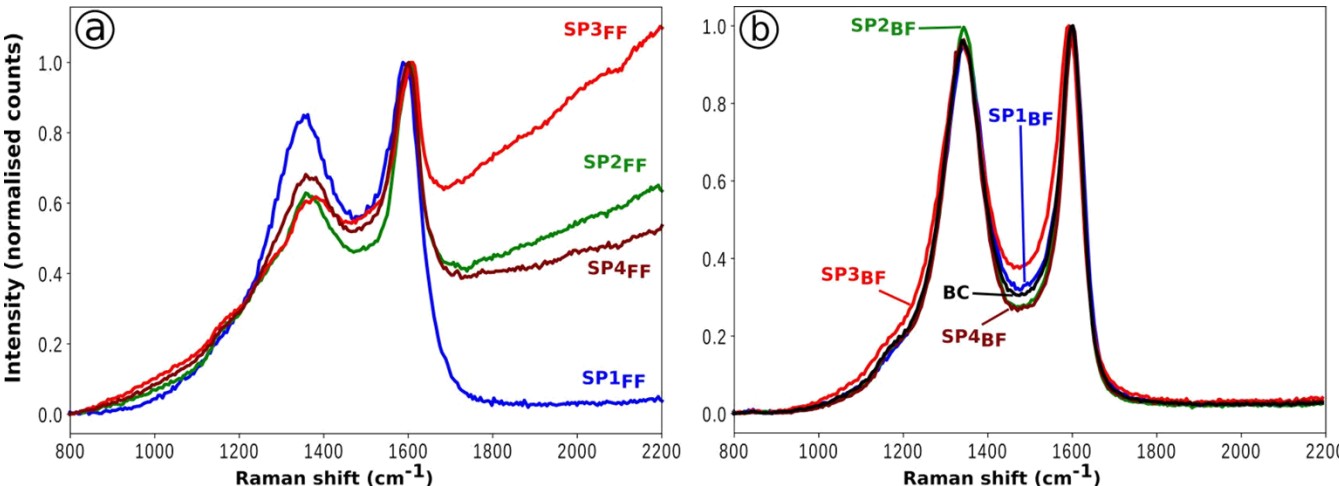

**Figure 10.** Raman spectra of SP1 (blue), SP2 (green), SP3 (red), and SP4 (brown) samples on Front Filter (a) and Back Filter (b). For all samples, plots are normalized to the maximum intensity of the G band. The spectrum of pure black carbon (BC) deposited on a QFF is plotted in black in panel (b) for comparison.

Both the fluorescence background (FB) and the soot Raman feature are observed to vary significantly with the set point (Fig. 10). The former refers to the baseline, whereas the latter refers to the two broad bands centered at 1356 $cm^{-1}$ and 1598 $cm^{-1}$, termed D (for defect) and G (for graphite), which correspond to Raman scattering involving $E_{2g}$ and $A'_{1g}$ symmetry, respectively (Ferrari and Robertson, 2000; Sadezky et al., 2005). FB is attributed to surface organic content (Cloutis et al., 2016) and is observed to decrease with increasing oxidation air flow for Front Filter samples ($FB_{SP3} > FB_{SP2} > FB_{SP4} > FB_{SP1}$). This trend is even clearer when the fluorescence slope fitted as a straight line between 800 and 2200 $cm^{-1}$ (Raman shift) is plotted against the total PAH signal determined in L2MS (Fig. 11). The linear fit ($R^2 = 0.992$) obtained in Fig. 11 reflects the good agreement between the FB and the measured trend in total PAH signal. For Back Filter samples, the FB signal is nonexistent and spectra resemble that of black carbon except in the "valley" region (i.e. in between the two soot peaks).

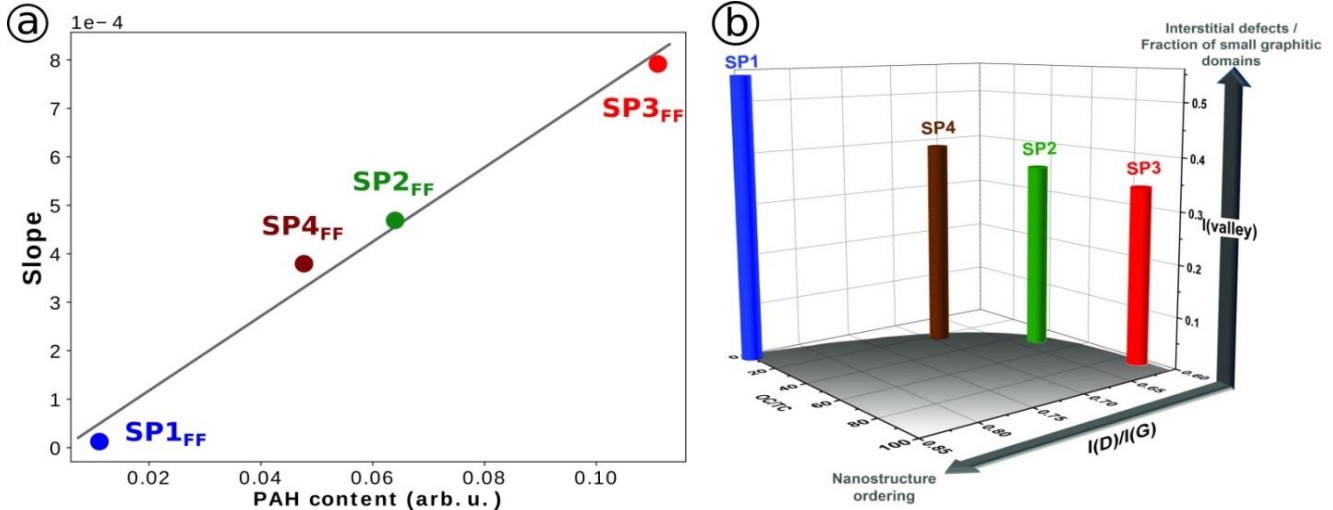

**Figure 11.** (a) Fluorescence slopes extracted from Raman spectra of SP1$_{FF}$ - (blue), SP2$_{FF}$ (green), SP3$_{FF}$ (red), and SP4$_{FF}$ (brown) plotted against total PAH content measured in L2MS (in terms of ion signal) in Front Filter samples, (b) 3-D bar plot showing the evolution of the intensity of the "valley" between the two D and G bands and that of the I(D)/I(G) height-ratio with organic content (OC/TC ratio) in Front Filter samples.

Two conclusions can be drawn from these observations. First, when comparing fluorescence signals of Back and Front Filter samples to PAH content, we can further refine our definition of organic content. Fluorescence is not just related to the total PAH signal, although this is a good marker of organic content. If it were, fluorescence would also be observed for Back Filter samples, in accordance with their relatively high gas-phase PAH contents. The lack of fluorescence signal on Back Filters, whose chemical composition is dominated by small PAHs, suggests that the fluorescence can be attributed mainly to non-volatile PAHs in the particulate phase, even though the heaviest mass detected in L2MS (*m/z* 546) is still small to expect fluorescence with a 514 nm excitation wavelength (Mercier et al., 2019). Consequently, the interaction of PAHs with one another or with the PM surface seems to trigger the fluorescence excited at 514 nm. Such perturbative effect (e.g. aggregate formation) on the luminescence has been observed in solutions (Nakagawa et al., 2013).

Information on soot nanostructure ordering can be derived from the I(D)/I(G) height-ratio, whose increase in intensity reflects a higher degree of order for soot made of crystallites ($L_a$) smaller than 2 nm (Ess et al., 2016; Ferrari and Robertson, 2000). The intensity of the valley region between the two peaks (1440–1540 cm$^{-1}$) provides insights into the presence of interstitial defects which may distort the lattice structure (Ess et al., 2016; Ferrari and Robertson, 2000; Sadezky et al., 2005) or into the occurrence of small graphitic domains (Parent et al., 2016). For comparison purposes, Raman spectra measured for Front Filters (Fig. 11a) have been subsequently baseline-subtracted and normalized to the G band before interpretation. Figure 11b shows the evolution of the I(D)/I(G) height-ratio and that of the intensity of the "valley" region with the OC/TC ratio for each set point. This 3D-bar plot confirms that the nanostructure order (I(D)/I(G) height-ratio) increases with decreasing OC content, i.e. the degree of order in the large polyaromatic network is higher under SP1 conditions (i.e. at higher oxidation air flow).

This is in agreement with the analysis of Ess et al. (2016) of soot produced at set points SP1, SP2, and SP3. Furthermore, a similar behavior with the oxidation air flow has been observed through an analogous Raman tracer (AD1/(AG+AD2)) involving the integrated band areas (Carpentier et al., 2012), from soot consisting of polyaromatic units poorly linked together (analogous to $SP3_{FF}$) to soot made of cross-linked structures with aliphatic bridges (analogous to $SP1_{FF}$). Figure 11b shows that the SP1 set point produces soot composed of a larger polyaromatic network (larger crystallite size) and a greater fraction of small interconnected graphitic domains (high "valley" intensity) as compared to $SP3_{FF}$. These results are in line with what has been previously observed for such miniCAST soot particles (Marhaba et al., 2019; Ouf et al., 2016), where the SP3 set point produced particles with very small crystallites (0.48–0.6 nm) and the greatest tortuosity/disordered structure (HRTEM) among all miniCAST samples.

## 4 Conclusions

Combustion by-products (PM and gas-phase) produced by a miniCAST generator are first separated and then characterized using a two-filter collection method and a multi-technique analytical/statistical protocol. Front and Back Filters thus generated are representative of the exhaust stream and are subsequently analyzed through first, an original L2MS technique featuring three ionization schemes, followed by SIMS, and last micro-Raman spectroscopy. The three-wavelength L2MS scheme is employed in our study to target specific classes of compounds. On the one hand, we evidence the presence of aliphatic compounds and specific fragment ions (118 nm), and on the other hand, we can focus on aromatic species (266 nm). Aromatic species were detected in all mass spectra (L2MS and SIMS). When combined with advanced statistical methods (PCA), mass spectrometry datasets revealed how different all samples were. Based on the PAHs classification of Bari et al. (2010), we were able to discuss aromatics distribution across Front and Back Filters in terms of volatile (1–2 rings), semi-volatile (3–4 rings), and non-volatile PAHs (larger than 4 rings). We determined that PM is essentially sampled on Front Filters, whereas the dominant compounds trapped on all Back Filters were volatile PAHs regardless of the combustion conditions. The good separation between the two phases confirmed the high particle collection capability of QFF Front Filters. PCA revealed that distinct amounts of volatile compounds were present in samples produced with different combustion parameters. Specifically, changes in oxidation air flow conditions in the miniCAST resulted in notable changes in the mass distribution for both Front and Back Filters. L2MS results at 266 nm indicated that low oxidation air flow conditions (SP2 and SP3) produced more semi-volatile and non-volatile compounds in the exhaust stream. The addition of quenching gas ($N_2$) in the miniCAST combustion conditions (SP4) lessened the difference between Front and Back Filters which featured more homogeneous mass spectra. Complementary micro-Raman spectroscopy analyses not only confirmed the relationship between the underlying fluorescence and the total PAH signal determined by mass spectrometry, but also identified as non-volatile the nature of PAHs involved in the fluorescence and detected in the particulate phase. Finally, all analyses confirmed the validity of total PAH signal as a proxy for the organic content (OC) commonly detected in thermo-optical measurements (Bescond et al., 2016; Yon et al., 2015). Accordingly, the total PAH signal measured by mass spectrometry was observed to decrease with increasing oxidation air flow conditions in the miniCAST. The two-filter collection method in conjunction with our multi-technique

analytical/statistical protocol allows the study of particles deposited on filters while preserving their predominant chemical properties, which makes possible comparisons with those determined by in-situ techniques (e.g. Ouf et al., 2016).

*Data availability.* The data presented here can be provided on request to the contact author.

*Supplement link (will be included by Copernicus)*

*Author contribution.* YC, CP, AF, CI, and CF conceptualized and built the sampling system and defined the methodology;
LDN, YC, RI, CI, GL, and CP performed the sample collection; LDN and JAN (SIMS), DD and MV (L2MS), RI, JAN and CP (Raman) performed the analysis and data reduction; LDN, YC, DD, MV, JAN, CP, and CF interpreted the results and wrote the original draft with additional contributions from other co-authors: IKO, AF, CI, MZ, and BC. JY, ET, CI, YC and CF provided funding and access to experimental infrastructure and organized the sampling campaign. All co-authors reviewed and approved the manuscript.

*Competing interests.* The authors declare that they have no conflict of interest.

*Acknowledgements.* This work was supported by the French National Research Agency (ANR) through the PIA (Programme d'Investissement d'Avenir) under contract ANR-10-LABX-005 (CaPPA – Chemical and Physical Properties of the Atmosphere), the MERMOSE project sponsored by DGAC (French national funds), the European Commission Horizon 2020 project PEMs4Nano (H2020 Grant Agreement #724145), and the CLIMIBIO project via the Contrat de Plan État-Région of
the Hauts-de-France region. J.A.N. acknowledges the financial support of Horiba Scientific. The collection campaign was financed by GDR Suie (GDR CNRS 3622). The authors acknowledge N. Nuns for his support in acquiring the SIMS data.

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
