# Peer review of "S1 L2MS spectra"

_Atmospheric Measurement Techniques, 2019_

## Referee Comment (RC1) · Anonymous Referee #1 · 7 Oct 2019

General Comments

In this manuscript the authors describe results of an experimental study of the chemical composition organic gases and particles sampled from the exhaust of a model combustion system (miniCAST) that is commonly used for conducting studies of combustion products under well controlled conditions. Particles were first collected on a quartz fiber filter and then gases that passed through the filter were collected on a second filter that was coated with black carbon to enhance adsorption efficiency. The filters were analyzed directly using a number of techniques, including two-step (desorption and ionization) laser mass spectrometry, secondary ion mass spectrometry, and

micro-Raman spectroscopy. Mass spectra were analyzed using principal component analysis. The major focus of the analysis was soot particles and PAHs, which are an important class of compounds formed by incomplete combustion, some of which are known to be carcinogenic. The experiments and data analysis seem to have been very carefully done, and the interpretation of the results was also very thorough. The manuscript provides a useful demonstration of the types of information that can be obtained on combustion products using this two-filter sampling method and suite of analytical instruments. I think it will be suitable for publication in AMT after the following comments are addressed.

Specific Comments

1. It is well established that quartz fiber filters (such as the front filter used to collect particles) adsorb organic vapors quite well. Was anything done in this study to evaluate the effect of this on the results and data interpretation? If not, then I suggest some vapors of standard PAHs with a range of volatilities be sampled and analyzed using this system. Alternatively, if others have conducted such studies then the authors could review the results of that work and discuss its consequences for the sampling and measurement approach employed here.

2. I found the Results and Discussion section rather challenging to read, due primarily to the somewhat monotonous style in which each observation was described in detail and then a possible explanation was provided. This made it difficult for me to differentiate important observations from minor ones. Although this made for a very thorough presentation, I'm not sure that readers will get the important take-away messages until they read the Conclusions (which may be all they choose to fully read). I suggest that the authors make a greater effort to emphasize the major points in each section of the manuscript, and perhaps eliminate some of the discussion that is mostly just minor observations with speculative explanations.

Technical Comments

Line 46: It seems unlikely that reviews published in 2011 and 2014 cover advances made over the last decade. Sentence should be reworded.

---

## Author Comment (AC1) · 31 Oct 2019

**Specific comments**

1) It is well established that quartz fiber filters (such as the front filter used to collect particles) adsorb organic vapors quite well. Was anything done in this study to evaluate the effect of this on the results and data interpretation? If not, then I suggest some vapors of standard PAHs with a range of volatilities be sampled and analyzed using this system. Alternatively, if others have conducted such studies then the authors could review the results of that work and discuss its consequences for the sampling and measurement approach employed here.

[Figure]

**Authors' response to specific comment 1):**

We thank the reviewer for his/her suggestion of discussing the consequences of the adsorption of organic vapors onto the filters in our two-filter sampling system and measurement approach. We will make sure that this possibility is acknowledged and briefly explain why we are confident that this phenomenon does not affect our results in a revised version of our manuscript. Please find our detailed response below.

The reviewer is right that the adsorption of gaseous hydrocarbons onto filters was investigated in numerous works. Specific studies assessing the ability of various filters to sorb gaseous organic species have even been carried out [e.g. 1-5]. They show that filters can collect organic vapors in addition to particulate matter. The efficiency of the adsorption of such vapors depends on a number of factors, including sampling duration and gas flow through the filter. In our study we used two filters placed in series in the exhaust line of a miniCAST, i.e. a bare quartz fiber filter (front filter) followed by a second black-carbon-covered quartz fiber filter (back filter). By doing so, we observed after a short sampling time (20 min) a clear partitioning where the particulate matter is essentially found on the first filter, while organic vapors (i.e. polycyclic aromatic hydrocarbons (PAHs) of different masses) are found condensed either onto both filters or just onto the back filter depending upon the masses of the PAHs.

Prior to sampling, spectra of neat quartz fiber filters (QFFs) have been recorded for both L2MS and SIMS measurements. The thermal treatment applied to the back filter proved to be very efficient at limiting the adsorption of aromatic species as evidenced when comparing the mass spectra of the quartz fiber filters before and after exposure to the exhaust. Moreover, the masses retained from our front and back filters to perform statistical analyses do not include specific masses associated with the substrate. Therefore, none of our deduction from statistical analysis is impacted by the possible condensation of organic vapors.

During our experiment, adsorption of organic vapors on the filters could affect the

chemical characterization of the particulate matter if we were using an analytical technique probing the bulk of the sample, i.e. from the sample surface all the way down to the filter. For instance, it is true that the adsorption of gaseous organic compounds onto filters is a potential source of errors in measurement when determining the mass of collected particles and the concentration of certain species/organic carbon (OC) in the particulate phase with a thermo-optical method [3-5]. However, in this work, we use instead two-step laser mass spectrometry (L2MS). L2MS is a surface characterization technique; since the laser penetration depth at $\lambda_d$=532 nm is only a few nm (orders of magnitude smaller than the average particle size in the studied regimes) only species present on the surface of the particulate matter are desorbed and analyzed. Therefore, if organic vapors are adsorbed onto the filter, they should not induce any measurement artifact when analyzing the particulate matter. This does not mean that organic vapor did not condense onto the particulate matter. We acknowledge and mention in the main text of our article that particulate matter likely consists of an adsorbed layer of organics onto an elemental carbon core. However, our experiment has not been designed to identify for certain whether heavy PAHs (>4 rings) are part of the particulate matter (chemisorbed or physisorbed) or "free" in the gas phase, insofar as they are both concomitantly present in the exhaust line and that we cannot avoid the fact that heavy PAHs may condense along with/onto the particulate matter when the latter is trapped by the first filter.

References:

1. Chase R., Duszkiewicz G., Richer, J., Lewis D., Maricq M., and Xu N., "PM Measurement Artifact: Organic Vapor Deposition on Different Filter Media," SAE Technical Paper, 2004, 2004-01-0967, 1-11.

2. Cotham W. E., and Bidlemant T. F., "Laboratory Investigations of the Partitioning of Organochlorine Compounds between the Gas Phase and Atmospheric Aerosols on Glass Fiber Filters", Environ. Sci. Technol., 1992, 26, 469-478.

3. Hartt K. M., and Pankow J. F., "High-Volume Air Sampler for Particle and Gas Sampling. 2. Use of Backup Filters To Correct for the Adsorption of Gas-Phase Polycyclic Aromatic Hydrocarbons to the Front Filter", Environ. Sci. Technol., 1994, 28, 655-661.

4. James J. Schauer, Michael J. Kleeman, Glen R. Cass, and Bernd R. T. Simoneit, "Measurement of Emissions from Air Pollution Sources. 2. C1 through C30 Organic Compounds from Medium Duty Diesel Trucks", Environ. Sci. Technol., 1999, 33 (10), 1578- 1587.

5. Mader B. T., and Pankow J. F., "Gas/Solid Partitioning of Semivolatile Organic Compounds (SOCs) to Air Filters. 3. An Analysis of Gas Adsorption Artifacts in Measurements of Atmospheric SOCs and Organic Carbon (OC) When Using Teflon Membrane Filters and Quartz Fiber Filters", Environ. Sci. Technol., 2001, 35(17), 3422-3432.

**2) I found the Results and Discussion section rather challenging to read, due primarily to the somewhat monotonous style in which each observation was described in detail and then a possible explanation was provided. This made it difficult for me to differentiate important observations from minor ones. Although this made for a very thorough presentation, I'm not sure that readers will get the important take-away messages until they read the Conclusions (which may be all they choose to fully read). I suggest that the authors make a greater effort to emphasize the major points in each section of the manuscript, and perhaps eliminate some of the discussion that is mostly just minor observations with speculative explanations.**

**Authors' response to specific comment 2):**

We will follow the reviewer's comment and improve the Results and Discussion section of our article in a revised version where we will outline the most important observations for each paragraph and remove extended descriptions/discussions about less important points that currently disrupt the thread.

**Technical comment**

**1) Line 46: It seems unlikely that reviews published in 2011 and 2014 cover advances made over the last decade. Sentence should be reworded.**

**Authors' response to technical comment 1):**

The sentence on line 46 will be reworded in our revised version. We will additionally replace "decade" by "decades" and add a more recent reference.

---

## Referee Comment (RC2) · Anonymous Referee #2 · 9 Nov 2019

The present paper examines the separation and the speciation of organic compounds in the particle and gaseous phases emitted during combustion processes. The laboratory experiments were conducted using a CAST burner. The authors proposed a two-filter sampling method to collect particulate matter on quartz fiber filters while the gas phase passed through the first filter and was then trapped on a second filter covered by black carbon. The samples are then analyzed using several techniques as two-step laser mass spectrometry (L2MS), secondary ion mass spectrometry (SIMS), and micro-Raman spectroscopy.

The experimental section and data analysis have been carefully carried out and fully

described. The manuscript is suitable to publication in AMT after revision.

General comments:

Collection system and experimental set-up. The collection system poses some serious problems. Positive artefacts are well known on quartz filters. Some of the filters were highly charged (PF1, PF2, PF3). We could expect that a kind of "soot cake" is formed on the front filters increasing the filtration efficiency and causing more adsorption of the gaseous compounds onto the deposited particles (see Fig. 1 images front filters).

In fig. 2 front and back filters mass spectra (L2MS) are presented. The m/z 202 ion partition on both the gas and particle phases (for SP1, SP2, SP4 but not for SP3) even though the signal is very high in the mass spectrum of the particle phase it almost not visible in the gas phase. How is it possible if for the other SPi it did partition? So why the partitioning of m/z 202 is so different for SP3 with respect to the other?

Another issue is the dilution system and the temperature in the sampling line. Low dilution at room temperature will enhance adsorption of relatively light PAHs onto the particles, while once in the atmosphere high dilution will alter this partitioning. This aspect has not been addressed.

It is relatively common, when working on filter samples, to remove the gaseous phase using denuders. This test has not be done apparently, and it would have been a easy test to verify the presence of organic volatile of the front filter.

Environmental relevance. The authors claim that their approach allows to identify distinct surface chemical compositions of aerosols discriminating semi-volatile and non-volatile polycyclic aromatic hydrocarbon (PAH) contents as a function of the combustion process. However modern engines are equipped with after-treatment devices that highly alter the exhaust emissions (oxidation of hydrocarbons and particle bound compounds). So in which way these results can be extrapolated to nowadays diesel engine emissions?

Quantification issues: since the author states that using three different ionization wave-lengths, it is possible to target various classes of compounds and to reach sub-fmol limit of detection, e.g. for PAHs (Faccinetto et al., 2008, 2015). So why the results are not present in a quantitative way?

Preparation of the filters. Prior to sampling the authors prepared the filters by heating them at $150°C$ for 16 hours. It seems to be a relatively low temperature with respect to the common procedure found in the literature for quartz filters (often up to $400°C$).

Poor English and paper structure. The paper is far too long and should be reduced in length and simplified. We need to understand what is important and not and to get few strong messages expressed in a clear and synthetic way. This paper is written as a scientific report or a student PhD thesis. I strongly suggest to revise completely the paper and possibly make nicer figures and plots.

PAC analysis. The added value was "to highlight variation and patterns in a data set, and in this case was used to reveal the differences in 210 chemical composition of the samples, and in particular between (i) Front and Back Filters and (ii) miniCAST set points." I can only partly agree to the added value of PCA analysis nevertheless the discussion is far too long. Please revise it.

---

## Author Comment (AC2) · 7 Dec 2019

We would like to thank the referee for his/her careful reading of the manuscript and the detailed comments that were provided. We think we addressed all the referee's concerns and we edited our manuscript when further details or clarifications were needed.

**1) Collection system and experimental set-up. The collection system poses some serious problems. Positive artefacts are well known on quartz filters. Some of the filters were highly charged (PF1, PF2, PF3). We could expect that**

[Figure]

**a kind of "soot cake" is formed on the front filters increasing the filtration efficiency and causing more adsorption of the gaseous compounds onto the deposited particles (see Fig. 1 images front filters).**

Authors' response to comment 1):

The referee explains that the trapping of particulate matter on the first filter could subsequently increase the filtration efficiency and therefore cause more gaseous compounds to adsorb onto the particles. While the referee's comment is true, another artefact is also reported in the literature: species originally adsorbed on the PM can also desorb during sampling and be retained on the back filter, which would lead to an overestimation of the gas-phase fraction (Paolini et al., 2017). We acknowledge and mention in the main text of our article that particulate matter likely consists of an adsorbed layer of organics onto an elemental carbon core (e.g. abstract l.19 or p.4, l. 117: "In the present study, the sampling line was designed to collect PM (including adsorbed species) on the Front Filter [. . .]"). Our experiment has not been designed to identify for certain whether heavy PAHs are part of the particulate matter (chemisorbed or physisorbed) or "free" in the gas phase, insofar as they are both concomitantly present in the exhaust line and that we cannot avoid the fact that heavy PAHs may condense along with/onto the particulate matter when the latter is trapped by the first filter. However, what our experimental set-up reveals is that different classes of compounds are almost solely found on each of the two filters. To highlight this observation, a "contrast function" defined as the $(S_{FF} - S_{BF})/(S_{FF} + S_{BF})$ ratio is represented for the 266 nm L2MS data in Fig. 1, where $S_{FF}$ and $S_{BF}$ are to the ion signals associated to a mass peak detected on the Front and the Back Filters. This representation clearly outlines that small aromatic species are found solely on Back Filters, whereas large PAHs are mostly on Front Filters. This suggests that the lightest species do not condense significantly onto the first filter and do pass through. As

what we referred to as semi-volatiles are found on both filters, it is indeed possible that PAHs included in that category are affected by an increased filtration efficiency of the first filter due the concomitant particulate matter adsorption (thus forming a thick soot deposit). This would result in a lower concentration of these PAHs on the back filter with respect to the front one, which would appear in our data as a lower absolute intensity of the corresponding mass peaks. However, this effect would not affect the covariance between mass peaks ($m/z$) as similar diffusion behaviors can be expected within SP1 and SP3 samples, for instance, because of similar soot porosity (e.g. the porosity of the soot "cake" deposited on silicon wafers for SP1 and SP3 set points were calculated to be about 98.1% and 97.4%, respectively, Ikhenazene et al., 2019). We therefore expect from a statistical standpoint that for each detected mass peak ($m/z$), the covariance will only negligibly be affected by diffusion.

Author's changes in the manuscript:

We first acknowledged this possibility in the article (page 4, line 118) by adding 2 sentences and the corresponding reference: "Note that particles build-up on the Front Filter could potentially increase its filtration efficiency and consequently trap PAHs that would rather pass through if the thickness of the PM collected on the Front Filter was not that high. Alternatively, species originally adsorbed on the PM can also be desorbed during the sampling and be retained on the back filter, which would lead to overestimate the gas-phase fraction (Paolini et al., 2017). However, our results will show that if this is the case, only specific PAHs of intermediate volatility are impacted by this phenomenon. In addition, this effect would not affect our statistical analysis (e.g. the covariance between mass peaks ($m/z$)) as similar diffusion behaviors can be expected within SP1 and SP3 samples, which exhibit similar soot porosity (e.g. the porosity of the soot material deposited on silicon wafers for SP1 and SP3 set points were calculated to be about 98.1% and 97.4%, respectively, Ikhenazene et al.,

2019). We therefore expect, from a statistical standpoint that for each given $m/z$, the covariance will only negligibly be affected by diffusion."

In addition, we introduced a figure and its description to demonstrate that some species are present specifically on either the Front Filter or the Back Filter, showing that the mentioned artefact is not dominant in our two-filter system for the non-volatile and volatile species, respectively (Fig. 4 in the main text and attached in copy to this Authors' response). The text page 9 line 261 reads now: *"The various CAST set points exhibit different PAH mass distributions on their Front and Back Filters, which likely relates to the different volatility properties of PAHs and probably affects their subsequent trapping on Front and Back Filters. Distinct volatility properties have been observed in the past on particles originating from wood combustion by Bari et al. (2010), who classified the PAHs on the basis of their number of aromatic rings resulting in the detection of three different PAHs categories. The authors classified the PAHs consisting of two aromatic rings as volatiles as they were mostly found in the gas phase, while those made of three and four rings were classified as semi-volatiles. PAHs comprising more than four rings were classified as non-volatile as they were observed in the PM in their study. Note that slightly different classes have also been defined in the literature (An et al., 2016; Elghawi et al., 2010; Sun et al., 2006). In our study, we largely found compounds consisting of one and two aromatic rings on Back Filters, while PAHs of $m/z$ $176 - 242$ were found on both Back and Front filters and those of $m/z \geq 252$ predominantly on Front Filters. Such PAH partitioning between Front and Back Filters is in line with the work of Bari et al. (2010). Similarly, we categorized the PAH distributions found on CAST samples into a volatile, semi-volatile, and non-volatile fraction (Fig. 3), where the volatile fraction encompasses here aromatic species made of one to two aromatic rings ($m/z$ $78 - 166$), the semi-volatile fraction comprises PAHs with a mass range of $m/z$ $176 - 242$, and the non-volatile fraction includes PAHs of $m/z \geq 252$. The boundaries of these intervals have been refined using the representation of Fig. 3 in which the "contrast function" defined as*

the $(S_{FF} - S_{BF})/(S_{FF} + S_{BF})$ *ratio is represented for the 266 nm L2MS data, where* $S_{FF}$ *and* $S_{BF}$ *are to the ion signals associated to a mass peak on the Front Filter and the Back Filter respectively. This representation clearly outlines that small aromatic species are found solely on the Back Filters, whereas large PAHs are mostly on the Front Filters."*

**2) In fig. 2 front and back filters mass spectra (L2MS) are presented. The** $m/z$ $202$ **ion partition on both the gas and particle phases (for SP1, SP2, SP4 but not for SP3) even though the signal is very high in the mass spectrum of the particle phase it almost not visible in the gas phase. How is it possible if for the other SPi it did partition? So why the partitioning of** $m/z$ $202$ **is so different for SP3 with respect to the other?**

Authors' response to comment 2):

The referee is concerned about specific behaviors observed for SP3. In the literature, various analyses converge towards the fact that the SP3 set point is distinct from the others in that i) the organic to total carbon ratio is higher (87% versus $\leq$ 47% for the other setpoints) − information that we confirmed with via our mass spectrometry measurements of the total PAH content (see Fig. 3 and Fig. 6), and ii) the crystallites of the particles produced in these conditions are significantly smaller and form a distinct disordered arrangement exhibiting many carbon edges (Bescond et al., 2016; Marhaba et al., 2019; Ouf et al., 2016; Yon et al., 2015). Such smaller crystallites suggest that SP3 may undergo nucleation and growth processes different from those of the other set points, thus leading to a different chemical composition for both particulate and gas phases.

The partitioning of SP3's $m/z$ $202$ ion between the Front and Back Filters is different from that of the other CAST set points. Specifically, the referee points out that, according to our L2MS mass spectrum, $m/z$ $202$ is almost exclusively found on the Front Filter in contrast to what is observed for SP1, SP2, and SP4. The partitioning of $m/z$ $202$ between FF and BF can be explained by thermodynamic and chemical considerations. First, it is useful to recall that mass spectrometry provides information about the mass of detected species (i.e. the chemical formula) but does not permit a direct determination of the structure of the detected molecules. In fact, for $m/z$ $202$ ($C_{16}H_{10}^+$), many different isomers (NIST Chemistry WebBook) can potentially contribute to the signal. The volatility of different isomers can vary, for instance the vapor pressure of fluoranthene is two times higher than that of pyrene (between 320 K and 390 K, Goldfarb and Suuberg, 2008). The relative contribution of different isomers to the detected signal highly depends on combustion conditions (i.e. CAST set points) and therefore on the chemistry. The high signal at $m/z$ $202$ observed for the SP3 Front Filter can be explained by a much higher contribution from isomers with a lower volatility (e.g. pyrene) produced in this regime. Second, it is clear that the chemistry in the reactive medium and the nature of the soot particles produced in the different set points of the miniCAST can lead to a large diversity in the relative chemical composition of the particulate phase and the gas phase.

Author's changes in the manuscript:

To address the referee's concern, we now discuss this particularity of $m/z$ $202$ in the main text of our article. We added on page 8, l. 253 the following discussion:

*"Literature data converge towards the fact that the SP3 set point is distinct from the others in that i) the organic to total carbon ratio is higher (87% versus $\leq$ 47% for the other set points), and ii) the crystallites of the particles produced in these conditions are significantly smaller and form a distinct disordered arrangement exhibiting many*

*carbon edges (Bescond et al., 2016; Marhaba et al., 2019; Ouf et al., 2016; Yon et al., 2015). Such smaller crystallites suggest that SP3 may undergo nucleation and growth processes different from those of the other set points, subsequently leading to distinct chemical compositions (e.g. different isomeric distributions) of the PM. The relative ion signals observed between the Front and Back Filters hence depend upon the relative volatilities and the response of the chemical compounds present on the samples to the 266 nm R2PI L2MS."*

**3) Another issue is the dilution system and the temperature in the sampling line. Low dilution at room temperature will enhance adsorption of relatively light PAHs onto the particles, while once in the atmosphere high dilution will alter this partitioning. This aspect has not been addressed.**

Authors' response to comment 3):

The referee is concerned that the low dilution/room temperature conditions in the sampling line as those used in our experiment induce the preferential adsorption of light PAHs on soot particles. For the sake of clarity, we would like to specify that the sole dilution system in our experimental setup is that of the miniCAST generator itself (dilution airflow 20 l min$^{-1}$, Fig. 1). We added a sentence on p.3, l.93 to make that fact clear: *"Note that the sole dilution system in our experimental setup is that*

Some studies are interested in probing the particulate matter in a chemical state as close as possible as that generated during the combustion process. In order to do so, it is necessary to avoid as much as possible any coating (PAHs, water) that could result from the adsorption of gaseous species during the collection, and accordingly, it is common to use additional dilution stages and denuders. In contrast, the goal of this study is to fully characterize the particulate and gas phase of combustion

by-products as they are emitted from the source, i.e. at the exhaust of a model soot generator without adding any dilution stage. We agree that the lower dilution of our sampling line plays on the partitioning of chemical species compared to systems having greater dilution factors. However, this phenomenon can be modeled (Lohmann and Lammel, 2004; Pankow, 1994) once the initial partitioning of chemical compounds (at the exhaust port) is known − information provided by this study. It is hence useful to characterize combustion by-products in the vicinity of the standardized source as it allows gathering data for subsequent simulations of the partitioning evolution in the atmosphere (at a different ambient temperature, air humidity, and pressure). In addition, Fig. 3 highlights that without using any extra dilution system, our two-filter approach is able to reproduce the separation commonly found in the literature regarding PAHs volatility properties, where the lightest are only found in the gas phase and do not condense on the PM collected on the Front Filters, and the non-volatile PAHs are predominantly found on the Front Filter.

Author's changes in the manuscript:

We added a sentence on p.3, l.93 to specify that we did not use an extra dilution system: *"Note that the sole dilution system in our experimental setup is that of the miniCAST generator itself (dilution airflow 20 l min$^{-1}$, Fig. 1)."*

**4) It is relatively common, when working on filter samples, to remove the gaseous phase using denuders. This test has not be done apparently, and it would have been a easy test to verify the presence of organic volatile of the front filter.**

Authors' response to comment 4):

As suggested by the referee, a commonly used approach for avoiding positive artifacts is the use of a denuder upstream of the quartz fiber filter to remove organic gases from the incoming air stream (e.g. Fitz, 1990). However, the removal of the organic gases also alters the gas-particle equilibrium and leads to a partial volatilization of the particle-bound organic phase, thus significantly changing the chemical composition of the particulate phase. Note that negative artifacts induced by the volatilization of species on the quartz fiber filter can also appear (Paolini et al., 2017). These artifacts can be accounted for, in particular for OC measurements of particulate matter, if the species that were vaporized are then trapped by an absorbing medium/filter (the total OC is the sum of OCs from two filters). However, since the efficiency of denuders is less than 100%, the gas phase will not only contain species that were vaporized from particles after the denuder but also a fraction of the original gas phase, and therefore the OC measured for the absorbing filter will be higher (positive artifacts). In other words, as the concentration of gas-phase organics is typically an order of magnitude higher than that of the particle-bound organic species, gas-phase material escaping the denuder can potentially create substantial positive sampling artifacts on the adsorbent backup filter (denuder breakthrough) (Subramanian et al., 2004). Sampling combustion by-products with the experimental system we propose, that is without a denuder, offers information about the unaltered gas and particulate phases, and therefore is more preferable to identify their presence on filters collected directly in the exhaust line, as is often the case in the literature (Crawford et al., 2011; Ess et al., 2016; Yon et al., 2015). Thermo-optical methods are often used to calculate the OC/TC ratio of CAST soot samples collected in the exhaust line. Our study provides information about the nature of the organic fraction and evidences the phase from which PAH molecules are most likely to originate. We do not exclude the condensation of the gas-phase on the particles once they are trapped by the Front quartz fiber filter, in fact, it is clear from the analysis of the two filters that PAHs classified as semi-volatile compounds and to a lesser extent as non-volatile compounds are found on both filters

with the largest fraction on the Front Filter, which could possibly be indicative of a preferential adsorption on the particles. This shows also that there are indeed PAHs of intermediate volatility on the particulate matter trapped on the Front Filter. However, it is also clear from our analyses that the lightest PAHs are not found on the Front Filter, which suggests that when gas-phase condensation occurs, it does not affect significantly the fraction constituted of the lightest PAHs.

**5) Environmental relevance. The authors claim that their approach allows to identify distinct surface chemical compositions of aerosols discriminating semi-volatile and nonvolatile polycyclic aromatic hydrocarbon (PAH) contents as a function of the combustion process. However modern engines are equipped with after-treatment devices that highly alter the exhaust emissions (oxidation of hydrocarbons and particle bound compounds). So in which way these results can be extrapolated to nowadays diesel engine emissions?**

Authors' response to comment 5):

As stated by the referee, it is true that modern certified engines are equipped with after-treatment systems (e.g. particle filters, catalytic strippers) that significantly change the chemical composition of both phases (particulate and gas phase). In a recent study we showed that a catalytic stripper that complies with European regulations for PMP systems ($> 99$ % removal of $\geq 30$ nm tetracontane particles) successfully removes the majority of particle-bound organic species (Focsa et al., 2019), and more specifically removed the volatile particles as well as the organic species from the surface of nonvolatile particles and subsequently increased the contribution of EC (carbon clusters) (Focsa et al., 2019). The proposed two-filter method can therefore be used to assess the efficiency of after-treatment systems by simultaneously measuring their impact on both particulate and gas phases. Moreover,

besides on-road vehicles (with certified engines), there are a large number of other sources of combustion by-products that do not have any after-treatment systems and for which our analyses are relevant. Aircraft jet engines, wood combustion stoves, biomass burning are just a few sources of combustion by-products whose exhaust is not subjected to an active after treatment system. Therefore, using a miniCAST soot generator operated with different parameters as a source of combustion by-products, which can mimic some of the physico-chemical properties of aircraft emissions for instance (Bescond et al., 2014; Marhaba et al., 2019; Moore et al., 2014), allows for potential real-world extrapolations of our results for combustion devices not equipped with after-treatment systems. The proposed method of sampling and characterizing concomitantly the particulate and gas phases can thus be extremely useful when evaluating the impact of these sources on the environment, as the gas/particulate portioning conditions the overall reactivity. As a perspective it would be very interesting test the efficiency of such devices especially when the engine starts.

Author's changes in the manuscript:

To account for the referee's comment we modified the original sentence, which now reads: *"The CAST (Combustion Aerosol Standard) generator is often chosen to produce combustion-generated particles as it is easy to implement for systematic laboratory experiments with, the fuel and oxidation air flows being easily modifiable, and hence enables the investigation of a variety of chemistries. Therefore, using a miniCAST soot generator operated with different parameters as a source of combustion by-products, which can mimic some of the physico-chemical properties of aircraft emissions for instance (Bescond et al., 2014; Marhaba et al., 2019; Moore et al., 2014), allows for potential real-world extrapolations of our results for combustion devices not equipped with after-treatment systems. The proposed method of sampling and characterizing concomitantly the particulate and gas phases can thus be extremely useful*

*when evaluating the impact of various sources (aircraft jet engines, wood combustion stoves, biomass burning) on the environment, as the gas/particulate partitioning conditions the overall reactivity. As it can simultaneously measure the particulate and gas phases, the proposed two-filter method can therefore be utilized to assess the efficiency of after-treatment systems, which are known to successfully remove the majority of particle-bound organic species (Focsa et al., 2019), and more specifically to remove the volatile particles and the organic species from the surface of nonvolatile particles while increasing the contribution of EC (carbon clusters) (Focsa et al., 2019)."*

**6) Quantification issues: since the author states that using three different ionization wavelengths, it is possible to target various classes of compounds and to reach sub-fmol limit of detection, e.g. for PAHs (Faccinetto et al., 2008, 2015). So why the results are not present in a quantitative way?**

Authors' response to comment 6):

The referee refers to the sentence on page 3, l. 77: "Using three different ionization wavelengths, it is possible to target various classes of compounds and reach sub-fmol limit of detection, e.g. for PAHs (Faccinetto et al., 2008, 2015)." Semi-quantitative results can be obtained with L2MS using external standards (synthetic soot), which consist of black carbon covered with chemical compounds whose surface concentration is well defined, and to which the signal collected in a sample can be compared (see Faccinetto et al., 2011, 2015) for a detailed description of the procedure). This approach can be used for PAHs exhibiting little to no fragmentation upon desorption and ionization, and is consequently more delicate to implement for aliphatic compounds, which undergo greater fragmentation after laser ionization (e.g. 118 nm). Moreover, this quantification step is complexified in the case of real soot by the presence of multiple PAH isomers for the same molecular formula. Such quantification study is out

of the scope of the present paper but will be in our priorities for future publications. Here we are only providing information about the chemical species present on the sample surface and their relative contribution with respect to their volatility. The limit of detection corresponds to the minimal desorbed amount of a specific compound that results in a detectable signal. Even though the limit of detection has been already measured for various PAHs (Faccinetto et al., 2011, 2015), this information cannot be used for quantification purposes. The limit of detection changes from isomer to isomer, therefore, for a reliable quantification, the relative contribution of all present isomers should be known − information that cannot be retrieved only from mass spectra. Since we are using desorption and ionization conditions akin to those described in Faccinetto et al. (2015), we assume that our detection limit is similar.

Author's changes in the manuscript:

To account for the reviewer's comment and dissociate the ideas of quantitative results and detection limit, we modified the sentence page 3, l. 77, which now reads: *"Using three different ionization wavelengths, it is possible to target various classes of compounds such as aromatic and aliphatic compounds. In addition, it is possible to reach a sub-fmol limit of detection for PAHs upon specific desorption and ionization conditions (Faccinetto et al., 2011, 2015)."*

**7) Preparation of the filters. Prior to sampling the authors prepared the filters by heating them at 150°C for 16 hours. It seems to be a relatively low temperature with respect to the common procedure found in the literature for quartz filters (often up to 400°C).**

Authors' response to comment 7):
The preparation procedure described above was primarily used to remove the pre-adsorbed species from the black carbon layer. The referee points out that the temperature of 150°C is lower than that commonly used (400°C). While this is true, all filters (bare quartz fiber filters and carbon-covered quartz fiber filters) have been analyzed in L2MS using the aromatic-selective 266 nm wavelength and in SIMS before they were placed in the exhaust line. As no signal could be evidenced in L2MS and only negligible signal was recorded in SIMS, we believe that our pre-treatment procedure prevented the adsorption of any unwanted compounds. In addition, it is worth recalling that the mass spectrometry techniques that we use are surface sensitive, which means that the signal we recover from Front and Back filters is that of the uppermost layer and thus for such thick samples our results are not affected by the composition of the substrate before sampling.

**8) Poor English and paper structure. The paper is far too long and should be reduced in length and simplified. We need to understand what is important and not and to get few strong messages expressed in a clear and synthetic way. This paper is written as a scientific report or a student PhD thesis. I strongly suggest to revise completely the paper and possibly make nicer figures and plots.**

Authors' response to comment 8):

We followed the referee's comment and improved the Results and Discussion section of our article in where we outlined the most important observations and shortened the extended descriptions. Specifically, we added 2-3 lines of conclusion at the end of each section (namely sections 3.1.1., 3.1.2., 3.2.1., 3.2.2.), we rewrote to a great extent sections 3.1.2 and 3.2.2. to outline the main findings, and we turned some

supporting materials to supplementary information (PCA for 157 nm and 118 nm). We think these changes improved the overall readability of our article. In addition, we greatly improved figures quality in order to make them clearer and nicer. Finally, the article will be read and corrected by a native English speaker before submission of the revised version. However, we do think that the actual length of our article is necessary to fully describe the comprehensive analyses that we performed. Author's changes in the manuscript: Please refer to our response to comment number 8) for the short description of our changes or to the main text for detailed information.

**9) PAC analysis. The added value was "to highlight variation and patterns in a data set, and in this case was used to reveal the differences in chemical composition of the samples, and in particular between (i) Front and Back Filters and (ii) miniCAST set points." I can only partly agree to the added value of PCA analysis nevertheless the discussion is far too long. Please revise it.**

Authors' response to comment 9):

We recently showcased the advantages of using advanced statistical techniques (Duca et al., 2019; Irimiea et al., 2018, 2019) to highlight subtle differences in the chemical composition of various samples. It is also very useful to confirm initial trends deduced from a descriptive approach of the mass spectra with a more quantitative approach such as that provided by the statistical analyses. However, to make the paper clearer and more synthetic, we did move PCA for 118 and 157 nm L2MS data to the supplementary information. PCA discussions for 266 nm L2MS and SIMS data have been rewritten to highlight the main conclusions and added values of these analyses.

Author's changes in the manuscript:

Subsections 3.1.2 and 3.2.2. have been rewritten as follows:

*3.1.2 Principal component analysis of L2MS spectra*

*In order to better discriminate the chemical composition of the various samples, particularly (i) the Front and Back Filters and (ii) the miniCAST set points, principal component analysis (PCA) was applied to mass spectra recorded for all three individual ionization wavelengths. A full description of this statistical method is provided in Sect. S2. Here, the covariance matrix was built from the integrated areas of all the detected peaks with a signal-to-noise ratio SNR > 3. The physical meaning of all derived principal components can be inferred from the contribution of the various molecular species to the loadings (see Sect. S2 and Fig. 5b and S3). By identifying the molecular families contributing to this variance, we can interpret the PCA score plots (Fig. 5) and grasp the nature of the subtle chemical differences between the samples. In L2MS data generated with 266 nm ionization wavelength, the scree and loading plots presented in Fig. 5b and S2a, respectively, 
[revised manuscript text omitted]

[Figure]

structure, J. Aerosol Sci., 101, 118–132, doi:10.1016/J.JAEROSCI.2016.08.001, 2016.

Crawford, I., Möhler, O., Schnaiter, M., Saathoff, H., Liu, D., McMeeking, G., Linke, C., Flynn, M., Bower, K. N., Connolly, P. J., Gallagher, M. W. and Coe, H.: Studies of propane flame soot acting as heterogeneous ice nuclei in conjunction with single particle soot photometer measurements, Atmos. Chem. Phys., 11(18), 9549–9561, doi:10.5194/acp-11-9549-2011, 2011.

Duca, D., Irimiea, C., Faccinetto, A., Noble, J. A., Vojkovic, M., Carpentier, Y., Ortega, I. K., Pirim, C. and Focsa, C.: On the benefits of using multivariate analysis in mass spectrometric studies of combustion-generated aerosols, Faraday Discuss., doi:10.1039/C8FD00238J, 2019.

Elghawi, U. M., Mayouf, A., Tsolakis, A. and Wyszynski, M. L.: Vapour-phase and particulate-bound PAHs profile generated by a (SI/HCCI) engine from a winter grade commercial gasoline fuel, Fuel, 89(8), 2019–2025, doi:10.1016/J.FUEL.2010.01.002, 2010.

Ess, M. N., Ferry, D., Kireeva, E. D., Niessner, R., Ouf, F. X. and Ivleva, N. P.: In situ Raman microspectroscopic analysis of soot samples with different organic carbon content: Structural changes during heating, Carbon N. Y., 105, 572–585, doi:10.1016/j.carbon.2016.04.056, 2016.

Faccinetto, A., Thomson, K., Ziskind, M. and Focsa, C.: Coupling of desorption and photoionization processes in two-step laser mass spectrometry of polycyclic aromatic hydrocarbons, Appl. Phys. A Mater. Sci. Process., 92(4), 969–974, doi:10.1007/s00339-008-4605-0, 2008.

Faccinetto, A., Desgroux, P., Ziskind, M., Therssen, E. and Focsa, C.: High-sensitivity detection of polycyclic aromatic hydrocarbons adsorbed onto soot particles using laser desorption/laser ionization/time-of-flight mass spectrometry: An approach to studying the soot inception process in low-pressure flames, Combust. Flame, 158(2), 227–239, doi:10.1016/j.combustflame.2010.08.012, 2011.

Faccinetto, A., Focsa, C., Desgroux, P. and Ziskind, M.: Progress toward the Quantitative Analysis of PAHs Adsorbed on Soot by Laser Desorption/Laser Ionization/Time-of-Flight Mass Spectrometry, Environ. Sci. Technol., 49(17), 10510–10520, doi:10.1021/acs.est.5b02703, 2015.

Fitz, D. R.: Reduction of the Positive Organic Artifact on Quartz Filters, Aerosol Sci. Technol., 12(1), 142–148, doi:10.1080/02786829008959334, 1990.

Focsa, C., Duca, D., Noble, J. A., Vojkovic, M., Carpentier, Y., Pirim, C., Betrancourt, C., P., Desgroux, Tritscher, T., Spielvogel, J., Rahman, M., Boies, A., Lee, K. F., Bhave, A. N., Legendre, S., Lancry, O., Kreutziger, P. and Rieker, M.: Multi-technique physico-chemical characterization of particles generated by a gasoline engine: towards measuring tailpipe emissions below 23 nm, Atmos. Environ., 2019.

Goldfarb, J. L. and Suuberg, E. M.: Vapor Pressures and Enthalpies of Sublimation of Ten Polycyclic Aromatic Hydrocarbons Determined via the Knudsen Effusion Method, J. Chem. Eng. Data, 53(3), 670–676, doi:10.1021/je7005133, 2008.

Ikhenazene, R., Pirim, C., Noble, J. A., Irimiea, C., Carpentier, Y., Ortega, I. K.,

Ouf, F.-X., Focsa, C. and Chazallon, B.: Ice Nucleation Activities of Carbon-Bearing Materials in Deposition Mode: From Graphite to Airplane Soot Surrogates, J. Phys. Chem. C, acs.jpcc.9b08715, doi:10.1021/acs.jpcc.9b08715, 2019.

Irimiea, C., Faccinetto, A., Carpentier, Y., Ortega, I. K., Nuns, N., Therssen, E., Desgroux, P. and Focsa, C.: A comprehensive protocol for chemical analysis of flame combustion emissions by secondary ion mass spectrometry, Rapid Commun. Mass Spectrom., 32(13), 1015–1025, doi:10.1002/rcm.8133, 2018.

Irimiea, C., Faccinetto, A., Mercier, X., Ortega, I.-K., Nuns, N., Therssen, E., Desgroux, P. and Focsa, C.: Unveiling trends in soot nucleation and growth: When secondary ion mass spectrometry meets statistical analysis, Carbon N. Y., 144, 815–830, doi:10.1016/J.CARBON.2018.12.015, 2019.

Lohmann, R. and Lammel, G.: Adsorptive and absorptive contributions to the gas-particle partitioning of polycyclic aromatic hydrocarbons: State of knowledge and recommended parametrization for modeling, Environ. Sci. Technol., 38(14), 3793–3803, doi:10.1021/es035337q, 2004.

Marhaba, I., Ferry, D., Laffon, C., Regier, T. Z., Ouf, F.-X. and Parent, P.: Aircraft and MiniCAST soot at the nanoscale, Combust. Flame, 204, 278–289, doi:10.1016/j.combustflame.2019.03.018, 2019.

Moore, R. H., Ziemba, L. D., Dutcher, D., Beyersdorf, A. J., Chan, K., Crumeyrolle, S., Raymond, T. M., Thornhill, K. L., Winstead, E. L. and Anderson, B. E.: Mapping the operation of the miniature combustion aerosol standard (Mini-CAST) soot generator, Aerosol Sci. Technol., 48(5), 467–479, doi:10.1080/02786826.2014.890694, 2014.

Ouf, F. X., Parent, P., Laffon, C., Marhaba, I., Ferry, D., Marcillaud, B., Antonsson, E., Benkoula, S., Liu, X. J., Nicolas, C., Robert, E., Patanen, M., Barreda, F. A., Sublemontier, O., Coppalle, A., Yon, J., Miserque, F., Mostefaoui, T., Regier, T. Z., Mitchell, J. B. A. and Miron, C.: First in-flight synchrotron X-ray absorption and photoemission study of carbon soot nanoparticles, Sci. Rep., 6, doi:10.1038/srep36495, 2016.

Pankow, J. F.: An absorption model of gas/particle partitioning of organic compounds in the atmosphere, Atmos. Environ., 28(2), 185–188, doi:10.1016/1352-2310(94)90093-0, 1994.

Paolini, V., Guerriero, E., Bacaloni, A., Rotatori, M., Benedetti, P. and Mosca, S.: Simultaneous Sampling of Vapor and Particle-Phase Carcinogenic Polycyclic Aromatic Hydrocarbons on Functionalized Glass Fiber Filters, Aerosol Air Qual. Res., 16(1), 175–183, doi:10.4209/aaqr.2015.07.0476, 2017.

Please also note the supplement to this comment:
https://www.atmos-meas-tech-discuss.net/amt-2019-275/amt-2019-275-AC2-supplement.pdf

———————————————

[Figure]

The plot (described in the figure below) shows a scatter plot with axis label $(S_{FF}-S_{BF})/(S_{FF}+S_{BF})$ on the y-axis ranging from -1.0 to 1.0, and m/z on the x-axis from 0 to 400. Right-side labels: 100% FF, 50% FF, 50% BF, 100% BF. Region labels: volatile, semi-volatile, non-volatile. Legend: SP1, SP2, SP3, SP4.

**Fig. 1.** "Contrast plot" representing the variation in PAH signal detected with L2MS at $\lambda i = 266$ nm for the 4 CAST set points.

---

## Author Comment (AC3) · 7 Dec 2019

More detailed and complementary answers to specific comment 1 can be found in the authors' response to specific comments 1, 3, and 4 of Referee #2. In particular, we specify the changes we wish to make in the main text to fully address this comment. Changes in the main text answering specific comment 2 are detailed in the authors' response to specific comments 8 and 9 of Referee #2. More specifically, we outlined the most important observations and conclusions of the Results and Discussion section. We rewrote to a great extent sections 3.1.2 and 3.2.2. to improved the overall readability of our article.